# Nuclear quantum effects in molecular liquids across chemical space

**Baris E. Ugur** ⓘ **& Michael A. Webb** ⓘ ✉

Nuclear quantum effects (NQEs) influence many physical and chemical phenomena, particularly those involving light atoms or occurring at low temperatures. However, their impact has been carefully quantified in few systems-like water-and is rarely considered more broadly. Here we use path-integral molecular dynamics to systematically investigate NQEs on thermophysical properties of 92 organic liquids at ambient conditions. Depending on chemical constitution, we find substantial impact across thermal expansivity, compressibility, dielectric constant, enthalpy of vaporization, and notably molar volume, which shows consistent, positive quantum-classical differences up to 5%; similar, less pronounced trends manifest as isotope effects from deuteration. Using data-driven analysis, we identify three features-molar mass, classical hydrogen density, and classical thermal expansivity-that accurately predict NQEs and facilitate understanding of how characteristics like branching and heteroatom content influence behavior. This work highlights the broad relevance of NQEs in molecular liquids, while also providing a conceptual and practical framework to anticipate their impact.

The quantum nature of nuclei influences every material and chemical process. Resulting nuclear quantum effects (NQEs) can arise as zero-point energy,[1] quantum tunneling[2,3], and quantization of proton energy levels. NQEs are generally expected to be substantial under certain conditions, such as processes occurring at low temperature or involving light nuclei. They have been found to be consequential in genetic stability of DNA[4,5], zeolite catalysts[6–9], superconductor materials[10,11], and enzymatic reactions[12–15]. NQEs also directly impact experimental studies that investigate or depend on isotope substitution[16–22]. Therefore, NQEs are important for many biological, chemical, and physical processes, even those containing heavier atoms or strong interactions[6,10,23–26], yet their extent and significance remain not well characterized for many systems.

Nuclear quantum effects are often deduced through equilibrium or kinetic isotope effects. In classical mechanics, kinetic isotope effects are expected but depend solely on the mass differences between isotopes; deviations from such results highlight quantum mechanical phenomena-such as nuclear tunneling. This is common in enzymatic reactions[13,27–32]; for instance, isotope substitution can alter the oxidation rate of yeast alcohol dehydrogenase by up to 33%.[15] In a very

different scenario, the conductivity of deuterated versus normal phosphoric acid differs by orders of magnitude, while classical expectations suggest a ratio of ca. 1.4[33]. Meanwhile, classical statistical thermodynamics does not predict isotope-dependent changes in equilibrium properties, yet isotope effects on such properties are widely observed[34–39]. For example, the solid-vapor isotope fractionation ratio between deuterated and normal water reaches values as high as 1.208[35]. Replacing $H_2O$ with $D_2O$ similarly impacts the structure and properties of biopolymer solutions[40,41], despite $D_2O$ being chemically equivalent to $H_2O$ under classical assumptions. The temperature-dependence of intramolecular isotopic equilibria has even been used to estimate dinosaur body temperatures via isotope-ratio mass spectrometry measurements on fossilized sauropod teeth[42]. The presence of such isotope effects signals the relevance of NQEs but does not fully characterize them.

Path-integral molecular dynamics (PIMD) enables explicit treatment of NQEs and calculation of isotope effects. This is in contrast to conventional classical MD, which treats nuclei as point particles evolving on a potential energy surface. Meanwhile, electronic structure methods, like density functional theory, provide a quantum

Department of Chemical and Biological Engineering, Princeton University Princeton, Princeton, NJ, USA. ✉e-mail: mawebb@princeton.edu

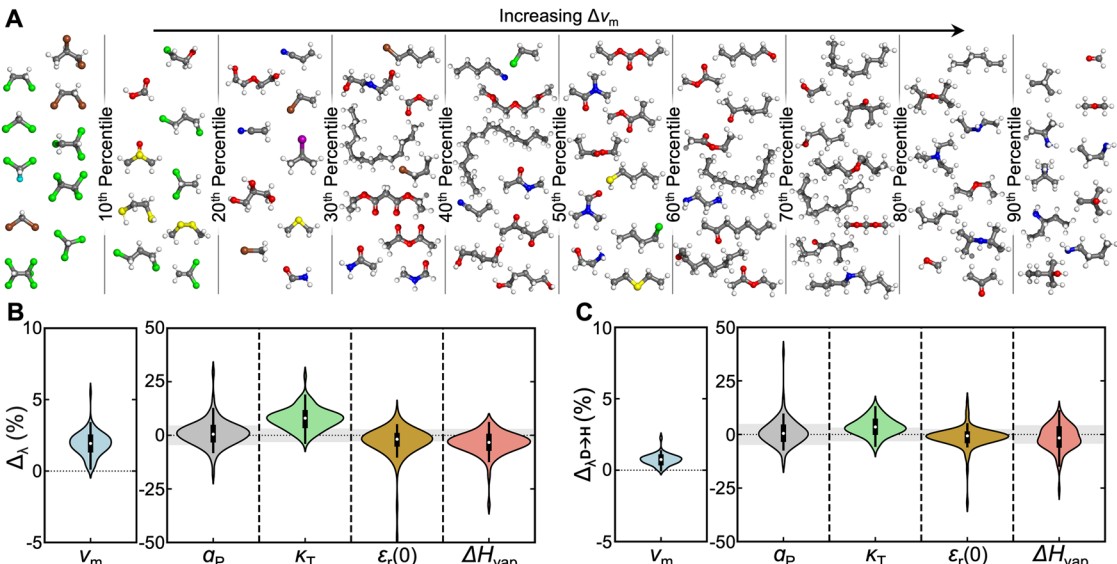

**Fig. 1 | Impact on thermophysical properties. A** Molecular structures of studied systems. Molecules are categorized into deciles based on $\Delta_{\nu_{m}}$. Within each decile, higher vertical position indicates higher $\Delta_{\nu_{m}}$. Atoms are colored as follows: carbon (gray), hydrogen (white), oxygen (red), nitrogen (blue), sulfur (yellow), chlorine (green), bromine (brown), iodine (purple). Structures were rendered with PyMOL (v3.1.0)[82]. The same data are shown with skeletal formula in Supplementary Fig. 1. **B** The magnitude of NQEs on molar volume, thermal expansion coefficient, isothermal compressibility, static dielectric constant, and enthalpy of vaporization of each system. **C** The calculated isotope effects on each property. The gray regions around $y = 0$ under each plot indicate the average standard error of the mean for the magnitude of NQEs on each property obtained from four independent simulations. The violin plots in (**B**, **C**) represent the distribution of $\Delta_{\lambda}$ and $\Delta_{\lambda^{D \to H}}$ values, with the white dot representing the median, the black box representing the interquartile range, and the inner lines representing the 1.5× interquartile range. The spread along the x-axis represents the density obtained from Gaussian kernel density estimation. All displayed properties are calculated at 298.15 K and 1 atm. Source data are provided as a Source Data file.

mechanical treatment of electronic degrees of freedom but generally rely on the Born-Oppenheimer approximation, under which the nuclei are held fixed or propagated classically. PIMD incorporates NQEs via a mathematical isomorphism, mapping each quantum nucleus to a classical ring polymer, thereby capturing quantum delocalization and thermal fluctuations by sampling in an extended phase space[43]. For water, PIMD has demonstrated how intramolecular zero-point motion and intermolecular tunneling affect properties like translational diffusion and orientational relaxation rates[44,45]. PIMD has also revealed the significance of NQEs in DNA base pairs, acetylene:ammonia co-crystals, and electrolyte transport in confined aqueous systems[46–48]. Nevertheless, in addition to the computational overhead of the extended phase space, PIMD can be practically limited by the availability of force fields that accurately represent the Born-Oppenheimer potential energy surface. Most conventional classical MD simulations typically use efficient force fields based on analytical equations. However, these force fields have parameters often calibrated to experimental data, implicitly including NQEs in uncontrolled ways, such that their use in PIMD risks "double counting" NQEs[44,49]. As a result, broad application of PIMD remains limited, and the role of NQEs across many systems and conditions is still largely unexplored.

Here, we characterize NQEs in 92 molecular liquids, spanning a wide organic chemical space, at ambient conditions. In particular, by comparing classical and path-integral MD simulations, we quantify NQEs on several thermodynamic properties including molar volume ($\nu_m$), thermal expansion coefficient ($\alpha_P$), isothermal compressibility ($\kappa_T$), static dielectric constant ($\varepsilon_r(0)$), and enthalpy of vaporization ($\Delta h_{vap}$). Equilibrium isotope effects on the same systems are also examined by simulating deuterated systems. This investigation spans a diverse set of chemistries-including amines, ethers, ketones, alcohols, alkanes, and more-and is enabled by the Topology Automated Force-Field Interactions (TAFFI) framework[50], an efficient analytical force field parameterized solely from quantum chemical calculations at the $\omega$B97X-D3/def2-TZVP level. Because TAFFI does not utilize

experimental calibration, it is well-suited for conducting PIMD across the numerous systems studied. We find a meaningful influence of NQEs for every substance and highlight key chemical features that correlate with NQEs using unsupervised and supervised machine learning. This reveals competing effects between system stability and hydrogen-atom densities on NQEs, which particularly explains trends related to hydrogen-bonding groups and molecular branching. By combining physics-based simulation and data-driven analysis, this work provides a deeper understanding of NQEs in common liquid organic systems and the conditions under which explicit consideration of NQEs may be needed.

## Results
### Varied effects across molecules and properties
To understand how NQEs manifest across chemical space, we simulate 92 organic molecules at ambient conditions. Of the 92 molecules studied, 87 were previously simulated using classical MD with various force fields[51], including TAFFI[50]. Benchmarking against experimental thermophysical properties showed that TAFFI compared favorably to other widely used empirical force fields in terms of reproducing experimental thermophysical properties across this set, which features amines, ethers, nitriles, ketones, alcohols, aldehydes, esters, amides, as well as some sulfur- and halogen-containing compounds. We further augmented this set with five n-alkanes ($C_9$, $C_{10}$, $C_{11}$, $C_{14}$, and $C_{15}$) to have representation of nonpolar hydrocarbons. The extent of NQEs is then quantified via

$$\Delta_{\lambda}(\%) = 100 \times \frac{\lambda^{PI} - \lambda^{cl}}{\lambda^{PI}} \qquad (1)$$

where $\lambda^{cl}$ and $\lambda^{PI}$ are the properties calculated via classical and path-integral MD simulations.

Figure 1 summarizes the results, with Fig. 1A qualitatively organizing all systems based on influence on molar volume, $\nu_m$. While it is clear and expected that heavy atoms like chlorine, bromine, and

sulfur lead to smaller NQEs, other relativistic organizing principles based on molecular constitution are less evident. More quantitatively, NQEs affect all systems and properties (Fig. 1B). Accounting for NQEs markedly increases molar volume by up to 5.5% and also increases $\kappa_T$ for most systems, with an average effect of nearly 8%. These systematic increases may signify weaker cohesive intermolecular interactions due to nuclear delocalization, in line with prior studies on water and other liquid organic systems[52–59]. In prior investigation featuring linear alkanes, the NQEs were found to decrease density by as much as 11%;[56] for water, NQEs induce a much smaller effect on the density, but its direction is consistent with our observations[56,60,61]. For reference, trends in density are opposite but similar in magnitude to molar volume, as analytically $\Delta_\rho = -\frac{\Delta_{v_\mathrm{m}}}{1-\Delta_{v_\mathrm{m}}} \approx -\Delta_{v_\mathrm{m}}$ (Supplementary Fig. 2). Quantum treatment of nuclei yields less consistent directional changes from classical results on the thermal expansion coefficient, enthalpy of vaporization and the static dielectric constant. For these properties, the average effect is small but also comparable to the statistical error in their calculation. These average effects are slightly positive for $\alpha_P$ and negative for $\varepsilon_\mathrm{r}(0)$ and $\Delta h_\mathrm{vap}$. Overall, these results indicate that NQEs can significantly and detectably influence certain thermodynamic properties across diverse molecular chemistries.

For the original TAFFI framework, all bonds are fit to purely harmonic functions, but anharmonicity in the potential energy surface-especially in bond-stretching modes-has been identified as a key factor influencing the magnitude of NQEs[44,62–64]. For example, in water, anharmonicity in the O-H bond contributes to competing effects: intermolecular zero-point energy and tunneling weaken hydrogen bonding, while intramolecular zero-point motion enhances the dipole moment and strengthens interactions, resulting in relatively small net NQEs[44].

To evaluate then whether the harmonic description in TAFFI may be responsible for the large NQEs relative to water, we reparameterized bonds involving hydroxyls, amines, and thiols (29 molecules) with an anharmonic Morse potential[44] based on expanded mode scans at $\omega$B97X-D3/def2-TZVP level of theory. We then evaluated $\Delta_{v_\mathrm{m}}$ with this anharmonic bond description and compared it to the harmonic results (Supplementary Fig. 3). Overall, we find that anharmonicity in bond-stretching slightly suppresses NQEs, though the overall magnitudes remain comparable to the harmonic description. We speculate that this mitigating effect is weaker in the organic molecules studied than in water, likely due to smaller dipole moment changes and correspondingly less enhancement of intermolecular interactions following introduction of anharmonicity. Intuitively, as most of the molecules in this study are larger than water, they are expected to exhibit smaller overall changes in dipole moment due to anharmonicity, as local bond distortions contribute less to the total molecular polarization. Although anharmonicity appears to have little impact for the molecules and conditions studied, its role in other phases, molecules, or force fields warrants careful consideration.

## Isotope effects versus nuclear quantum effects

Building on the preceding results that directly compare quantum and classical systems, we probe isotope effects of deuteration using

$$\Delta_{\lambda^{\mathrm{D}\to\mathrm{H}}}(\%) = 100 \times \frac{\lambda^{\mathrm{PI}} - \lambda^{\mathrm{PI,D}}}{\lambda^{\mathrm{PI,D}}} \quad (2)$$

where $\lambda^{\mathrm{PI,D}}$ is the property for a fully-deuterated system simulated via PIMD; deuterated systems are often expected to approximate classical behavior. The quantity $\Delta_{\lambda^{\mathrm{D}\to\mathrm{H}}}$ is equivalent to $1 - \frac{\lambda^\mathrm{H}}{\lambda^\mathrm{D}}$ where $\frac{\lambda^\mathrm{H}}{\lambda^\mathrm{D}}$ is often noted as the (experimentally accessible) equilibrium isotope effect. Trends in $\Delta_{\lambda^{\mathrm{D}\to\mathrm{H}}}$ (Fig. 1C) align directionally with $\Delta_\lambda$ (Fig. 1B) but with reduced magnitudes. However, beyond consistent shifts in molar

volume, the effect of deuteration is often within the range of statistical error. This change is most notable for $\kappa_T$, where the average effect shifts from 7.8% when comparing to purely classical systems to 3.5% when comparing to quantum mechanical but deuterated ones. This highlights that deuteration does not fully replicate a purely classical treatment. A system with a negligible isotope effect may still display significant NQEs, which partially cancel between the isotopically normal and deuterated systems.

## Key molecular determinants

We next identify key molecular features that influence the strength of NQEs in liquid systems. Given its consistent and statistically resolvable effects, in addition to $v_\mathrm{m}$ being a fundamental thermophysical property relevant to equations of state and other material properties, our discussion focuses on $\Delta_{v_\mathrm{m}}$, while analyses of other properties are included in Supplementary Fig. 4.

To visualize how NQEs vary across a broad chemical space, we apply dimensionality reduction from a high-dimensional molecular feature space to two dimensions using unsupervised machine learning. Specifically, we use the Uniform Manifold Approximation and Projection (UMAP) algorithm[65,66] on a dataset that includes our 92 investigated molecules and an extended set of 2874 molecules from the ChEMBL database[67,68]. Each molecule is initially represented by a 34-dimensional atom-type feature vector generated using the Merck Molecular Force Field (MMFF94)[69], and then UMAP constructs a neighborhood graph from the collection of such vectors, models pairwise relationships probabilistically, and optimizes a low-dimensional embedding that preserves both local and global structure. The resulting collective variables, $Z_1$ and $Z_2$, are non-linear transformations of the high-dimensional feature values, which obfuscates facile interpretation. Nevertheless, $Z_1$ and $Z_2$ define a coordinate space that can be easily visualized and for which chemical similarity trends with distance.

Within the broader chemical context of the ChEMBL database, we find that the simulated substances indeed span a wide chemical space (Fig. 2A). In terms of readily identifiable trends, chemical structures displaying lower $\Delta_{v_\mathrm{m}}$ are seemingly clustered in the $Z_1$-$Z_2$ space in the vicinity of molecules with generally higher molecular weights, such as chloroform. In contrast, molecules with larger $\Delta_{v_\mathrm{m}}$ are distributed across the manifold, indicating that diverse molecular features may enhance NQEs. This is further supported by the fact that different functional groups can be associated with population-level effects on $\Delta_{v_\mathrm{m}}$ (Supplementary Fig. 6). For instance, molecules with amines tend to exhibit higher $\Delta_{v_\mathrm{m}}$, and those containing heavier atoms exhibit lower $\Delta_{v_\mathrm{m}}$. At the same time, there is significant variation of $\Delta_{v_\mathrm{m}}$ within each population of functional groups, such that attributing NQEs purely to functional groups is difficult. This highlights the necessity to consider additional factors to ultimately understand the manifestation of NQEs.

To gain insight into what factors correlate with NQEs, we create a data-driven model for $\Delta_{v_\mathrm{m}}$ from simple descriptors, with the rationale that an effective model would highlight the importance of those descriptors. We find that a model using just three inputs-average atomic mass ($m_w$), hydrogen density ($n_\mathrm{H}^\mathrm{cl}$), and thermal expansion coefficient ($\alpha_P^\mathrm{cl}$)-accurately predicts $\Delta_{v_\mathrm{m}}$ (Fig. 2B). The coefficient of determination over all predictions is $R^2 = 0.881 \pm 0.002$. Here, predictions are made as $\hat{\Delta}_{v_\mathrm{m}} = \hat{T}_{v_\mathrm{m}} \alpha_P^\mathrm{cl}$, where $\hat{T}_{v_\mathrm{m}} = \mathrm{RF}(m_w, n_\mathrm{H}^\mathrm{cl})$ is the output of a random forest (RF) regressor trained to predict a defined quantity $T_{v_\mathrm{m}} \equiv (\alpha_P^\mathrm{cl})^{-1} \Delta_{v_\mathrm{m}}$, and the RF model is trained on all data except that of the molecule being predicted. We empirically find that this approach outperforms models formulated as $\hat{\Delta}_{v_\mathrm{m}} = \mathrm{RF}(m_w, n_\mathrm{H}^\mathrm{cl}, \alpha_P^\mathrm{cl})$, likely due to the large variability in $\alpha_P^\mathrm{cl}$ across molecules and its nature as a response function, unlike the more intrinsic descriptors $m_w$ and $n_\mathrm{H}^\mathrm{cl}$. As a point of possible interest, we note that $T_{v_\mathrm{m}}$ has units of temperature and, to leading order, can be interpreted as the temperature shift required for a classical system to match the molar volume of its quantum

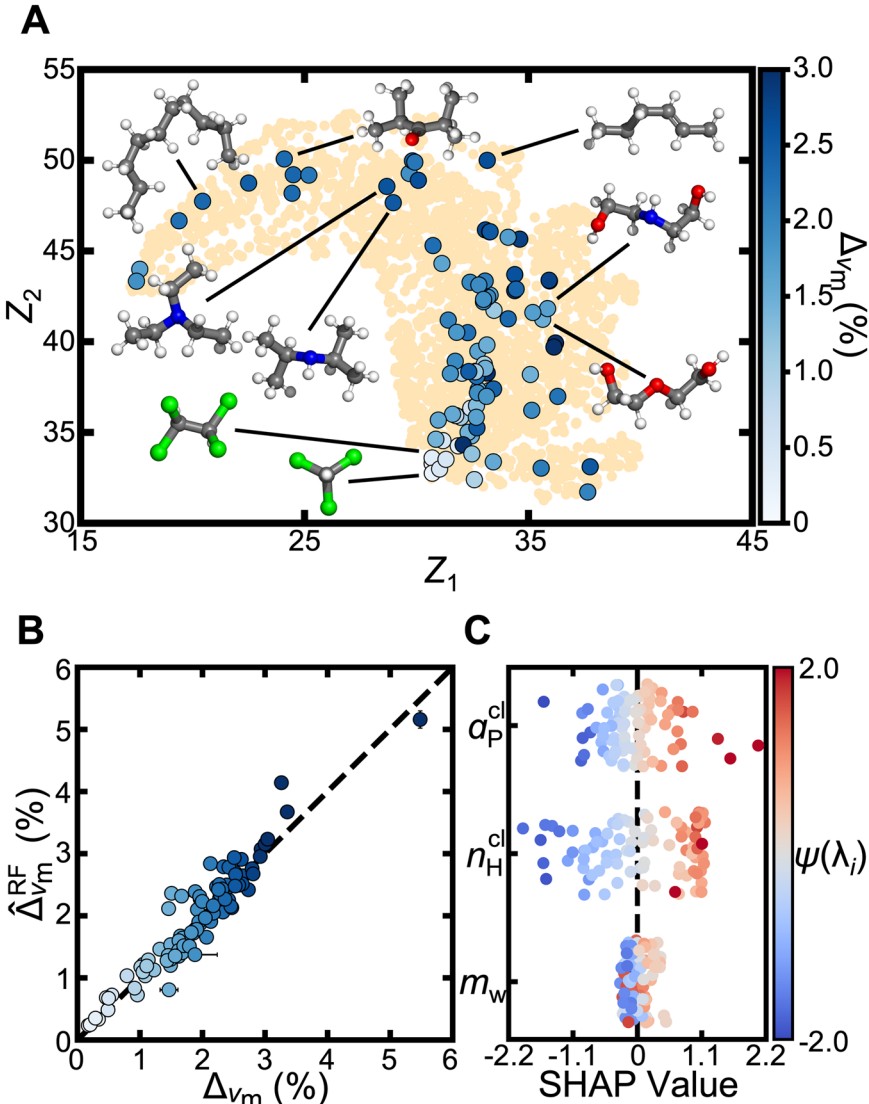

**Fig. 2 | Correlation of chemical features with the magnitude of NQEs. A** A two-dimensional manifold organization of the 92 organic molecules (blue, larger markers) and 2874 small molecules (tan, smaller markers) obtained from the ChEMBL database. The coordinates $Z_1$ and $Z_2$ are derived by applying the Uniform Manifold Approximation and Projection (UMAP) algorithm to a 34-dimensional input vector based on MMFF94 atom typing[69]. Inset lines highlight representative molecules from different regions of the manifold. **B** Comparison of predicted versus simulated effect of NQEs on molar volume ($\Delta_{v_m}$). Predicted values are obtained from a data-driven model with three chemically interpretable input features: average atomic mass ($m_w$), hydrogen density ($n_H^{cl}$), and thermal expansion coefficient ($\alpha_P^{cl}$); the latter two are determined from classical MD simulations. Error bars reflecting the standard error of the mean, determined using four independent simulations and from five different training and testing cycles for the data-driven model, are generally smaller than the symbol size. The markers (blue) use the same color scale as those in (**A**). **C** Impact of feature contributions based on Shapley Additive Explanations (SHAP) analysis. The position on the x-axis indicates the impact of each feature on the model output, and the marker color indicates feature value. Feature values ($\lambda_i$) are displayed following a Yeo-Johnson power transformation, denoted by $\psi$, to present data on similar scales[83]. For visual clarity, the SHAP data are shown over a restricted range; Supplementary Fig. 5 depicts the full observed range of data. Source data are provided as a Source Data file.

counterpart (i.e., $v_m^{cl}(T + T_{v_m}) \approx v_m^{PI}(T)$). This interpretation does not imply a true physical equivalence between the quantum system and a classical system at elevated temperature-a comparison that has been cautioned against as potentially misleading in prior work[70]. Across the molecules studied, $T_{v_m}$ ranges from a few to several tens of degrees (Supplementary Fig. 7).

We use Shapley Additive Explanations (SHAP) analysis[71] to attribute feature contributions to predictions made by our machine learning model. This technique assesses the impact of each feature by computing the difference in model output with and without its inclusion for all feature combinations. This analysis reveals some intuitive trends: abundance of hydrogens and

greater $n_H^{cl}$ enhance $\Delta_{v_m}$ (Fig. 2C). These effects reflect a general dependence on molecular composition, as systems with low mass-density or hydrogen-rich systems would generally imply greater sensitivity to NQEs due to the presence of lighter nuclei. However, $\alpha_P^{cl}$ correlates positively with $\Delta_{v_m}$. SHAP analysis also indicates that $\alpha_P^{cl}$ and $n_H^{cl}$ features have greater impact on model output than $m_w$, which has a non-monotonic relationship with $\Delta_{v_m}$. Although variable importance was established within the context of 92 organic molecules, it is useful to consider how these principles translate to a well-studied system, like water. Using classical simulation data from the q-TIP4P/F water model[44]

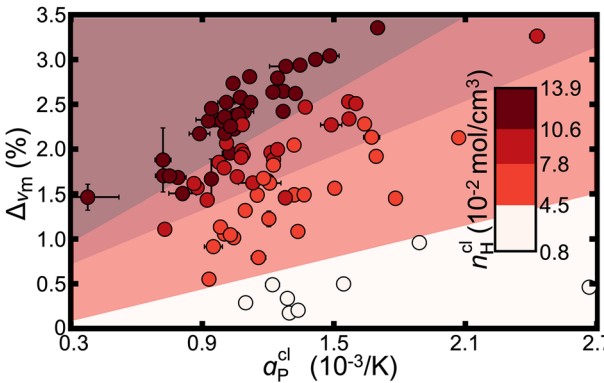

**Fig. 3 | Analysis of hydrogen density and thermal expansion coefficient.** Four groups of data are generated with $k$-means clustering algorithm on the respective hydrogen densities $n_H$; the marker colors indicate being within the range specified by the color bar. The colored regions are distinguished by support vector machine (SVM) margin lines. Error bars (black) represent the standard error of the mean obtained from four independent simulations. For visual clarity of trends, the data are shown over a restricted range; Supplementary Fig. 8 depicts the full observed range of data. Source data are provided as a Source Data file.

as input, the data-driven model predicts $\hat{\Delta}_{\nu_m}^{RF} = 0.47\%$, in good agreement with the simulation result $\Delta_{\nu_m} = 0.25\%$. Despite water being chemically distinct from the other molecules studied, the model captures the small impact on $\nu_m$ due to NQEs–an outcome largely driven by its low $\alpha_P$.

Overall, these results indicate a complex interplay between NQEs, fluid composition, and its stability that is yet predictable through simple properties readily computed with classical simulation.

### Competing effects of system stability and hydrogen density

At first glance, the trends for $n_H$ and $\alpha_P$ appear to reflect competing effects: higher $n_H$ suggests stronger intermolecular forces, which would typically reduce $\alpha_P$. To explore this trade-off, we leverage the dataset to create controlled comparisons by fixing either $n_H$ or $\alpha_P$ and examining the effect of the other on $\Delta_{\nu_m}$. This is accomplished through $k$-means clustering to group molecules with similar $n_H$ values and examining correlations of the groups with $\Delta_{\nu_m}$ (Fig. 3). Within groups with similar $n_H$, a decrease in fluid stability (indicated by higher $\alpha_P$) tends to enhance NQEs. Conversely, across groups with comparable stability (similar $\alpha_P$), an increase in $n_H$ (and thus more light nuclei) leads to stronger NQEs. Nevertheless, the overlapping $\alpha_P$ ranges across different $n_H$ groups (with some weak negative correlation) suggest that both properties facilitate predicting the magnitude of NQEs.

### Trends with hydrogen bonding and branching

To elucidate how $\Delta_{\nu_m}$ correlates with molecular constitution, we systematically compare groups of molecules varying in hydrogen bonding (Fig. 4A) and branching (Fig. 4B) and examine how key descriptors, $n_H$ and $\alpha_P$, are affected.

In Fig. 4A, comparing 1-bromobutane (i), 1-chlorobutane (ii), butane-1-thiol (iii), and butan-1-ol (iv) highlights the positive impact of hydrogen bonding on $n_H$ and $\Delta_{\nu_m}$. Among these molecules– which share the same number of heavy atoms, the same chemical topology, and comparable $\alpha_P$–the hydrogen-bonding thiol and alcohol exhibit higher $n_H$. However, $n_H$ and $\Delta_{\nu_m}$ do not scale monotonically with additional hydrogen-bonding groups. This is revealed by comparison of butan-1-ol (iv), butane-1,4-diol (v), pentane-1,5-diol (vi), and propane-1,2,3-triol (vii). Among these alcohols, increasing the number of hydrogen-bonding groups minimally affects hydrogen density but increases the stability of the fluid (decreases $\alpha_P$). Evaluation of

different hydrogen-bonding groups reveals similar trends (Supplementary Fig. 9) where the balance of $n_H$ and $\alpha_P$ elucidates the resulting $\Delta_{\nu_m}$. For example, relative to alcohols with similar $n_H$, amines display larger NQEs and associated larger $\alpha_P$.

In Fig. 4B, comparing multiple linear molecules and their branched analogues, we observe consistent impact on NQEs. In particular, the branched molecule exhibits larger $\Delta_{\nu_m}$ than the linear one. Across all four pairs, branching reduces $n_H$, but there is a coupled increase in $\alpha_P$. For these molecules, the net result is stronger NQEs in the branched molecules. We therefore suggest that NQEs, at least as manifested through $\Delta_{\nu_m}$, are likely to be more substantial in branched molecules owing to relatively diminished fluid stability. This observation is reminiscent of an expected trend: boiling points for branched molecules are typically lower than those of linear molecules of comparable molecular weight and composition.

### Detailed physics of linear versus branched systems

The conventional rationale for the trend involving boiling temperatures is that branched molecules have reduced surface area and less efficient packing, which weakens intermolecular forces and impacts the magnitude of NQEs. Intriguingly, the linear and branched molecules studied here exhibit comparable interaction patterns with respect to the number of hydrogen bonds, their strength, and average nearest-neighbor distances (Supplementary Fig. 10). This renders the importance of hydrogen-bonding groups on trends in $\alpha_P$, and thus their impact on $\Delta_{\nu_m}$, unclear.

Resolving interaction patterns as a function of distance between neighboring molecules, however, illustrates key differences between butan-1-ol with its branched analogue, 2-methylpropan-2-ol (Fig. 5). The distribution of hydrogen bonds (Fig. 5A) shows that butan-1-ol forms bonds over a broad range of intermolecular distances, while methylpropan-2-ol, with less conformational flexibility, forms bonds only at separations of approximately 4.6 Å. A further distinction emerges in energetics of molecular interactions within the fluid (Fig. 5B): 2-methylpropan-2-ol exhibits a pronounced minimum in interaction energy near the distance associated with hydrogen bonding, while butan-1-ol shows steadily diminishing attractive interactions with distance. These nuanced changes in intermolecular interactions due to branching result in a pronounced increase in $\alpha_P$, which tends to enhance $\Delta_{\nu_m}$. Notably, the minimum interaction energy for butan-1-ol occurs at a distance where few or no hydrogen bonds form. Additionally, butan-1-ol lacks a strong preference for relative molecular orientation, unlike 2-methylpropan-2-ol (Fig. 5C). These findings suggest that the cohesive energy of 2-methylpropan-2-ol relies more heavily on hydrogen bonding whereas butan-1-ol exhibits overall stronger dispersion forces. This comparison also serves to demonstrate the sensitivity of NQEs to subtle changes in nanoscale interactions.

## Discussion

Nuclear quantum effects (NQEs) are present in all physical systems, but their significance is rarely known beyond certain well-studied systems (e.g., water) or expected scenarios (e.g., low temperatures). In this study, we quantified the impact of NQEs at ambient conditions on various properties across 92 organic molecules spanning diverse chemical space. By comparing path-integral and classical simulations, we observed that NQEs tend to significantly and measurably enhance molar volumes and isothermal compressibilities. Effects on thermal expansion coefficients, dielectric constants, and enthalpies of vaporization lack clear general directional trends relative to the uncertainty of the calculations themselves, though individual systems can still be significantly impacted. Simulations of deuterated systems reveal similar but reduced effects, with isotope-induced changes generally within the statistical uncertainty, except for molar volume. This suggests deuteration often underestimates the full

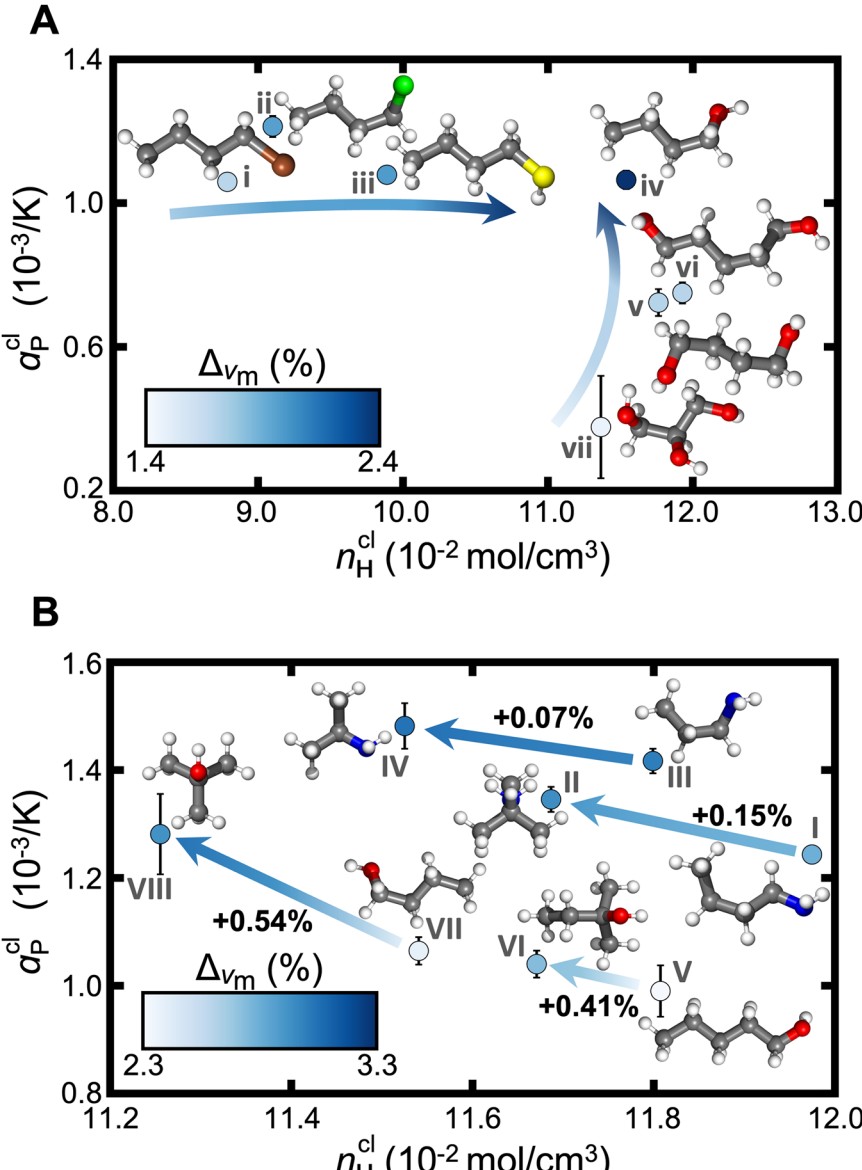

**Fig. 4 | Impact of hydrogen bonding and branching on the extent of NQEs.**
**A** Hydrogen densities and thermal expansion coefficients for systems of various hydrogen-bonding groups and counts. Visualized molecules are labeled as (i) 1-bromobutane, (ii) 1-chlorobutane, (iii) butane-1-thiol, (iv) butan-1-ol, (v) butane-1,4-diol, (vi) pentane-1,5-diol, (vii) propane-1,2,3-triol for reference. **B** Comparison between four pairs including linear molecules and their branched analogues of chemically similar structures. The impact of branching on the thermal expansion coefficient and hydrogen density is shown on the y- and x-axis, respectively. The marker color intensity denotes the magnitude of NQEs, indicated by the color bars and arrow directions in each panel. Visualized molecules are labeled as (I) butan-1-amine, (II) 2-methylpropan-2-amine, (III) propan-1-amine, (IV) propan-2-amine, (V) pentan-1-ol, (VI) 2-methylbutan-2-ol, (VII) butan-1-ol, (VIII) 2-methylpropan-2-ol. It should be noted that while the two color bars have different bounds, the range of the bounds is equal. Error bars (black) represent the standard error of the mean obtained from four independent simulations. Source data are provided as a Source Data file.

extent of NQEs, as a quantum treatment of deuterium still impacts properties.

We further identified physical properties that effectively correlate with the magnitude of NQEs. This was particularly demonstrated by a data-driven model with interpretable inputs of the average mass of composite atoms, hydrogen densities, and thermal expansion coefficients. We posit this as a pragmatic approach to anticipate the relevance of NQEs and the necessity of computationally intensive path-integral simulations. Beyond prediction, these property trends highlight key physical factors, some intuitive and others less so, affecting NQEs. An intriguing finding from the analysis is the competing effect between hydrogen density and fluid stability (linked to the thermal

expansion coefficient), where both factors enhance NQEs but are somewhat inversely correlated.

By exploiting the chemical diversity of our dataset, molecular-level insights were obtained by examining sets of molecules with variations in hydrogen content, hydrogen-bonding capabilities, and comparisons between linear and branched topologies. Grouping data based on functional groups revealed that NQEs are generally more significant in amines compared to ethers and alcohols, for example. Broadly, systems with high hydrogen density (and therefore light nuclei) but weaker interactions, leading to lower stability, exhibit the largest NQEs. This explains a potentially unintuitive result that systems with multiple hydrogen-bonding groups may not necessarily exhibit strong NQEs because the hydrogen bonding tends to enhance fluid

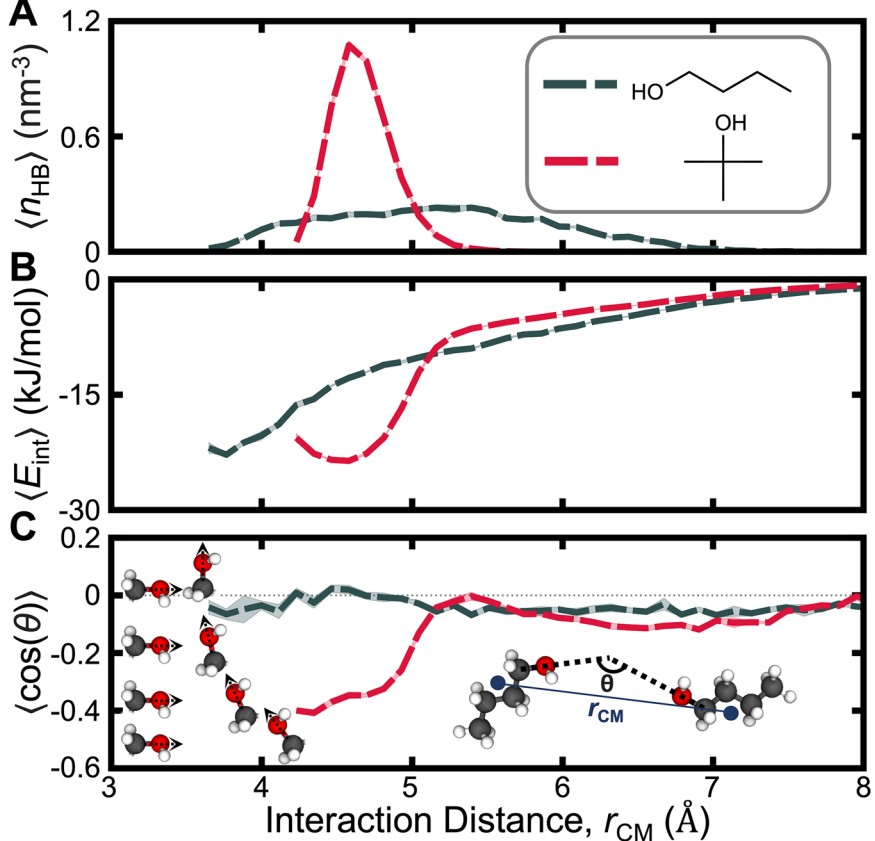

**Fig. 5 | Effect of branching on intermolecular interactions through comparison of linear butan-1-ol with branched 2-methylpropan-2-ol.** Ensemble averages of (**A**) the hydrogen-bond density, (**B**) pairwise intermolecular interaction energies, and (**C**) interaction orientations as a function of distance between molecules. The inset images of panel (**C**) illustrate calculations of the interaction distance and orientation; the renderings adjacent to the y-axis illustrate configurations consistent with the nearest tick marks. Error regions indicate the standard error obtained from four simulations for each bin. All molecular renderings were generated using the Visual Molecular Dynamics (VMD) software[84] (version 1.9.4). Source data are provided as a Source Data file.

stability, reflected in a smaller thermal expansion coefficient. By contrast, molecular branching has a general tendency to both reduce fluid stability and hydrogen density. This study advances our understanding of NQEs in commonly used substances, guiding experimental and computational approaches and providing a framework for when explicit treatment of NQEs is necessary to capture essential physical phenomena.

## Methods

### General simulation protocols

Systems were modeled with the quantum chemistry-based Topology Automated Force-Field Interactions (TAFFI) framework[50] with Waldman-Hagler mixing rules. The TAFFI framework was chosen for this study due to its efficiency as an analytical force field, its strict parameterization using quantum chemical calculations at the $\omega$B97X-D3/def2-TZVP level, and the generally good agreement for a range of thermophysical properties, comparing values calculated from classical simulations and those reported experimentally[50]. Real-space non-bonded interactions were truncated at 14 Å, unless otherwise noted. Long-range electrostatics calculations used the Particle Mesh Ewald (PME) algorithm; tail corrections were applied for Lennard-Jones interactions beyond the real-space cutoff, unless otherwise noted. All PIMD simulations used a ring-polymer bead count of $P = 32$. All simulations used a timestep of 0.5 fs. Constant-temperature conditions of 298.15 K were achieved using a Langevin thermostat for classical simulations and the path-integral Langevin equation (PILE) thermostat[72] for path-integral simulations; both utilize a friction coefficient of 1 ps$^{-1}$. Constant-pressure conditions at 1 atm were

achieved using a Monte Carlo barostat with an attempt frequency of 25 fs.

The force field in tandem with aforementioned settings enables accurate reproduction of liquid-phase experimental densities[51] for 87 systems with available empirical data (Supplementary Fig. 11). Benchmarking of classical and PIMD simulation results also shows that path-integral treatment of most systems brings predicted molar volumes closer to experiment. To gain quantitative insight, we apply a one-tailed paired t-test on the differences with experiment and report $p = 4.8 \times 10^{-17} < 0.05$, indicating that PIMD results are systematically closer to experiment. These results, along with good alignment between force-field intramolecular normal-mode frequencies and DFT results, suggest that TAFFI can effectively represent and discriminate the chemically diverse set of systems in this study.

Property calculations and analyses were performed for four independent classical and path-integral MD simulations of each molecular system. Each simulation contained the minimum number of molecules to reach 5000 atoms, initialized with random positions and orientations within a simulation cell of $V = 50$ nm$^3$. The initial configuration was subjected to energy minimization with an energy tolerance of 10 kJ/mol followed by 0.1 ns simulation in the microcanonical ensemble. The system was then equilibrated in the canonical ensemble at 400 K for 0.2 ns and cooled to 298.15 K over 0.8 ns with a linear temperature schedule. Subsequently, systems were equilibrated in the isothermal-isobaric ensemble at 1 atm for 3 ns followed by a production run of 10 ns, during which thermodynamic data were collected, and system configurations were saved every 10 ps. Simulations of q-TIP4P/F water systems followed a similar procedure with 1000 water

molecules and with a real-space non-bonded cutoff distance of 10 Å. The above procedure was found to be insufficient to equilibrate $C_{14}$ and $C_{15}$ systems. For these, initial structures were generated using a 100 ns classical simulation in the isothermal-isobaric ensemble at 298.15 K and 1 atm, followed by 3 ns of additional equilibration and 10 ns production runs using either classical MD or PIMD. These protocols were sufficient to achieve adequate energy conservation and effectively converge the system density (see Supplementary Figs. 12 and 13).

All simulations were performed using the GPU implementation of the OpenMM 7.7.0 software package[73]. PIMD simulations were performed on NVIDIA A100 and P100 GPUs, with mean simulation times of 46.5 and 89.3 h, respectively, in contrast with classical MD simulations of 1.7 h on A100 GPUs. Total simulation time throughout the study was 55,383 GPU hours.

### Gas-phase simulations
Single-molecule simulations of each system were used to calculate the gas-phase energy to obtain $\Delta h_{vap}$. A molecule was placed in a large box with a volume of 125 $nm^3$ and simulated in the microcanonical ensemble for 1 ns, followed by the canonical ensemble at 298.15 K for 0.4 ns using an Andersen thermostat with a collision probability of 10 $ps^{-1}$. For these simulations, all pairwise interactions were evaluated in real-space and restricted to within 15 Å to ensure that there are no interactions with periodic images. Linear and angular momenta of each system were removed, the particle velocities were rescaled, and the potential energies were saved every 1 ps.

### Analysis of system properties
Molar volumes were calculated using

$$\upsilon_m = \frac{M}{\langle \rho \rangle} \tag{3}$$

where $M$ is the molar mass of the molecule and $\langle \rho \rangle$ is the average (mass) density.

Thermal expansion coefficients, $\alpha_P$, were calculated using

$$\alpha_P = \frac{1}{\langle V \rangle} \left( \frac{\partial V}{\partial T} \right)_P \tag{4}$$

where $\left( \frac{\partial V}{\partial T} \right)_P$ was numerically approximated using average volumes from simulations at 293.15, 298.15, and 303.15 K. The additional simulations at 293.15 and 303.15 K were performed using the equilibrated configurations from 298.15 K. The systems were equilibrated in the isothermal-isobaric ensemble at for 0.5 ns, followed by 1.5 ns production runs where the system volume was saved every 1 ps.

Isothermal compressibilities, $\kappa_T$, were calculated using the fluctuations in the system volume using

$$\kappa_T = \frac{\langle V^2 \rangle - \langle V \rangle^2}{k_B T \langle V \rangle} \tag{5}$$

where $k_B$ is the Boltzmann constant. Similarly, static dielectric constants, $\varepsilon_r(0)$ were calculated using

$$\varepsilon_r(0) = \frac{1}{P} \sum_{i=1}^{P} 1 + \frac{4\pi}{3} \frac{\langle \mathbf{M_i^2} \rangle - \langle \mathbf{M_i} \rangle^2}{V k_B T} \tag{6}$$

where $\mathbf{M}$ is the total dipole moment of the simulation box ($P = 1$ for classical MD simulations).

Molar heats of vaporization, $\Delta h_{vap}$, were calculated using

$$\Delta h_{vap} = \frac{1}{P} \sum_{i=1}^{P} E_i^{(g)} + k_B T - \frac{1}{P} \sum_{i=1}^{P} E_i^{(l)} \tag{7}$$

where $E^{(\upsilon)}$ is the total internal energy per mole (including both potential and kinetic energy) in phase $\upsilon$, either liquid or gas. This expression

assumes the ideal gas approximation and that the vapor-phase molar volume is much greater than that of the liquid. Often, the kinetic energy contributions in the gas and liquid phases are expected to cancel, allowing $\Delta h_{vap}$ to be approximated from potential energies alone[51]. However, in PIMD simulations, quantum kinetic energy-arising from harmonic coupling of systems in the extended phase space-can differ between phases. Therefore, we retain all energy contributions in the calculation for simplicity.

### Unsupervised Learning
Unsupervised learning via Uniform Manifold Approximation and Projection (UMAP) dimension reduction was performed using the Python umap-learn package (version 0.5.3), with MMFF94 atom types to describe the 34-dimensional feature space. The MMFF94 atom types were obtained using the rdkit Python package (version 2022.9.3)[74]. The UMAP analysis includes additional small organic molecules obtained from the ChEMBL database. Molecules containing rings, net charge, fluorine atoms, or more than two double or triple bonds were excluded, resulting in 2874 molecules. For the UMAP hyperparameters, the analysis used 100 for the size of local neighborhood, 1.0 for the minimum distance between embedded points and Euclidean distances. The $k$-means clustering algorithm to categorize the systems into four groups based on their $n_H$ was generated using the scikit-learn Python package (version 1.0.2) with default hyperparameters[75].

### Supervised Learning
Supervised learning via random forest regression used the scikit-learn Python package. Model inputs included $m_w$ and $n_H$ from classical MD simulations. Model training and assessment used a leave-one-out split. Each molecule was tested by training the model on the other 91 molecules. This process was repeated for all 92 molecules. The reported $R^2$ value and its associated error were calculated from five iterations with different random seeds. Each random forest model used 20 decision trees and required two samples per split. Shapley Additive Explanations (SHAP) analysis was performed using the shap Python package (version 0.44.1) on each of the three input parameters for the data-driven model.

The LIBSVM implementation of C-Support Vector Classification algorithm[76,77] in scikit-learn Python package (version 1.0.2) was used to generate support vector machine boundary lines for four groups of molecules based on their $n_H$. A linear kernel type and a regularization parameter of 1.0 were chosen as the algorithm hyperparameters.

### Analysis of molecular interactions
All interaction analyses were performed and averaged over configurations sampled every 1 ns from four independent simulations of each molecule. The interaction distance, $r_{CM}$, was defined as the distance between the centers of mass of two molecules. The array of interaction distances was discretized into 50 bins between 3.3 and 9 Å. Hydrogen-bond densities ($n_{HB}$) were defined as the number of hydrogen bonds in each bin per volume. Hydrogen bonds were defined using geometric criteria with a cutoff distance of 3.6 Å between the donor and the acceptor and a cutoff angle of 150 degrees between the three participating atoms. These calculations were facilitated using the MDAnalysis Python library (version 2.1.0)[78,79]. Interaction energies, $E_{int}$, were computed as the sum of all pairwise non-bonded energy contributions between two molecules. Interaction orientations, $\cos(\theta)$, were calculated based on the angle ($\theta$) between the bond vectors adjoining the carbon to the oxygen on the alcohol.

### Impact of force field
As previously discussed, TAFFI is used because it relies purely on quantum chemical calculations for its parameterization and computational efficiency; however, we acknowledge that the reported magnitude of NQEs could differ based on the underlying force field. To gain

some insight into the sensitivity of NQEs, classical and PIMD simulations using two different force fields–the all-atom optimized potentials for liquid simulations (OPLS-AA) force field and the Open Force Field (OpenFF, version 2.0.0 with unconstrained bonds)–were also performed and analyzed for a subset of 11 molecules, following the procedures described in Methods.

Relative to the TAFFI results, OPLS-AA and OpenFF consistently find larger $\Delta v_{\mathrm{m}}$ across all systems (Supplementary Fig. 15). It is important to note that OPLS-AA and OpenFF include empirical adjustments, potentially inherited from prior developments, making them strictly unsuitable for PIMD simulations. As a result, this analysis is merely suggestive rather than definitive regarding differences from alternative representations of the Born-Oppenheimer potential energy surface. However, the results do not indicate significant exaggeration of effect sizes from TAFFI. Future work could explore variations arising from entirely different interaction models, such as machine learning potentials.

## Data availability

The processed data generated in this study[80] have been deposited in the Zenodo database under accession code https://doi.org/10.5281/zenodo.15236880.

## Code availability

Analysis scripts and source code used in this research are publicly accessible at https://github.com/webbtheosim/chem-space-nqes, including instructions for reproducing the results. The code[81] has been deposited in the Zenodo database under accession code https://zenodo.org/records/15465591.

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

## Acknowledgements

B.E.U. and M.A.W. acknowledge support from the National Science Foundation under Grant No. 2237470 (M.A.W.). The authors also acknowledge support from the "Chemistry in Solution and at Interfaces" (CSI) Center funded by the U.S. Department of Energy through Award No. DE-SC0019394 (M.A.W). Simulations and analyses were performed using resources from Princeton Research Computing at Princeton University, which is a consortium led by the Princeton Institute for Computational Science and Engineering (PICSciE) and Office of Information Technology's Research Computing. These resources include a GPU-based computing cluster purchased with support from the National Science Foundation (Grant No. NSF-MRI: OAC-2320649) (M.A.W.). We thank Hang Zhang for his valuable assistance with quantum chemistry mode scans for force field parameterization.

## Author contributions

M.A.W. and B.E.U. designed research; B.E.U. performed research; B.E.U. and M.A.W. analyzed data; B.E.U. and M.A.W discussed results; B.E.U. and M.A.W. wrote the paper; B.E.U. and M.A.W. edited the paper.

## Competing interests

The authors declare no competing interests.
