## [Transparent Peer Review file · Nature Communications]

Nuclear Quantum Effects in Molecular Liquids Across Chemical Space

Corresponding Author: Professor Michael Webb

Version 0:

Reviewer comments:

Reviewer #1

(Remarks to the Author)

In this work, Ugur and Webb perform path-integral computer simulations of a large number (87) of liquids/molecules at ambient conditions. The aim of this work is to identify small molecules for which the thermodynamic properties of the corresponding liquid (nuclear volume, thermal expansion coefficient, isothermal compressibility, and dielectric constant, enthalpy of vaporization) are/are not sensitive to nuclear quantum effects (as well as isotope H D substitution effects). It is found that nuclear quantum effects (NQE) at ambient conditions (1 bar, 298 K) are relevant over a wide range of molecules, inducing up to a 5%-change in the corresponding liquid molar volume. Using machine learning, the authors show that four thermodynamic properties obtained from classical MD simulations are good indicators of those liquids for which the target thermodynamic properties are/are not affected by NQE.

The question raised in this work is important and the work is timely. NQE have been studied in a few molecular liquids, particularly, water. While in this case NQE can be relevant, it is not clear the impact of NQE on other small organic molecules. The manuscript is clear and well-written. While I find that the manuscript could be published in Nat Comm, there are a few points that should be addressed.

1) That NQE should be relevant at low temperatures is expected. However, in this work the NQE are observed at ambient conditions (1 bar and 298 K). This may be somewhat surprising since even for water, the thermodynamic properties are barely sensitive to NQE [e.g., Eltareb et al. Phys Chem Chem Phys. 23, 6914-6928 (2021); Stolte, et al., J Phys Chem Lett. 15, 12144-12150 (2024)]. Can the authors comment on this? Have the authors studied water as well? (it is hard to tell from Fig. 1A).

Related to this question, how relevant is "a $\leq 5\%$ change in molar volume"? Can the authors provide an example/comparison showing the relevance of a 5%-change in molar volume? (eg, how would the density change of the liquids?)

2) It is not clear how/why the 87 molecules studied were chosen. Is there any chemical feature or protocol followed to select them? Even when 87 molecules is a relevant number of molecules, it is a small sample of chemical space (eg, in Fig. 2A, it seems that the region of low-Z1 is not well-covered by the selected 87 molecules).

3) The use of machine learning to identify molecules prompt to NQE at ambient conditions is interesting. Unfortunately, the results are obscured by the numerical/statistical algorithms and lack of details. In page 4, the UMAP algorithm is used to classify the molecules in the Z1-Z2 plane. The algorithm should be described briefly. The collective variables Z1 and Z2 should be defined. (Fig. 2A). Similarly, the SHAP algorithm should also be described briefly in page 6.

4) Can the authors use their statistical results to identify some chemistry principle (e.g., by pointing to chemical groups) that give rise to NQE? Is/isn't there a correlation between the chemical groups and the presence of NQE in the studied molecules? The topic is briefly addressed in pages 5-6 in the context of "hydrogen bonding and branching" but it should be expanded.

5) In Fig.1B and 1C, what is the meaning of the colored cloud in the panels? I understand that the dispersion of the cloud along the y-axis represents the dispersion of the NQE effects $\Delta\lambda$. However, what is the dispersion of the cloud along the x-axes? Are units needed in the x-axes? Do the x-axes represent normalized properties?

Reviewer #2

(Remarks to the Author)

In this manuscript, Ugur and Webb using path integral molecular dynamics (PIMD) simulations to examine nuclear quantum effects (NQE) across a wide range of molecular liquids at ambient conditions, finding that NQEs are significant across the range of liquids. By computing thermodynamic quantities and performing further analysis using both classic techniques and machine learning, the authors identify physically meaningful trends that can be interpreted in terms of intuitive and chemically meaningful information. The complex data is presented in a clear and intuitive manner that helps identify trends and the major points being made. Overall, this is an interesting paper that should have broad appeal, and I recommend publication after the authors consider a few questions/comments.

- 1.) I think the comment "a null deuteration effect does not necessarily indicate negligible NQEs" is important. Nothing to do here, just want to point out that this is a useful statement. On a side note, the sentence needs a period at the end.
- 2.) On line 135, the authors mention excluding methanal, but prior to this, they do not point out that methanal is the outlier. Could you please point this out before this sentence? It is also not clear to me why having a low boiling point would cause problems. Could you please explain this point in a little more detail?
- 3.) Line 143 needs "to" between "due" and "the", and a space between "substances" and n_H^c .
- 4.) The work seems to focus on polar liquids and/or liquids with specific interactions. Do you have any expectations for how nonpolar liquids like alkanes or fluorinated hydrocarbons would fit into the new perspective provided here? For example, would alkanes not follow the trends involving n_H ?
- 5.) Line 554 has a space in the middle of the word "simulations" that should be removed.

Reviewer #3

(Remarks to the Author)

I enjoyed reading this paper, which explores local and global molecular indicators of quantum nuclear effects (NQEs) in molecular liquids. The analysis is sophisticated, employing state-of-the-art machine learning approaches. Their method successfully captures intuitive indicators of NQEs, such as those associated with the presence of light atoms. However, beyond these expected trends, I did not find the results entirely convincing. Some observed trends may even contradict previous studies. For example, Ref. [1] demonstrates that stronger hydrogen bonds enhance NQEs in the Ubbelohde effect, whereas this paper suggests that weaker interactions lead to stronger hydrogen bonding. Could this discrepancy arise because the force field used lacks anharmonicity?

Additionally, the paper is highly computational in focus, which may limit its appeal to experimentalists. I have two technical comments that the authors should consider, especially if submitting to a computationally oriented journal:

NQEs are known to be significantly modulated by anharmonic effects, which the authors do not account for [1,2,3]. At a minimum, this limitation should be explicitly discussed.

The expression for the enthalpy of vaporization is incorrect, as it assumes that the kinetic energy cancels between the liquid and gas phases.

[1] <https://www.pnas.org/doi/10.1073/pnas.1016653108>

[2] <https://journals.aps.org/prl/abstract/10.1103/PhysRevLett.117.115702>

[3] <https://pubs.acs.org/doi/10.1021/acs.jctc.9b00596>

Reviewer #4

(Remarks to the Author)

The manuscript by Ugur and Webb reports on a high-throughput study of nuclear quantum effects in 87 organic molecular liquids at ambient conditions using path integral molecular dynamics simulations. An exhaustive analysis of NQEs on several properties of the liquid (density, compressibility, thermal expansion, etc.) is carried out using state-of-the-art supervised and unsupervised statistical learning approaches, which reveal general trends. In systems with high hydrogen density and lower stability, NQEs have a higher impact on their properties. These observations provide useful indications to assess whether treating NQEs by PIMD is necessary to study certain systems. Additionally, the simulations indicate that NQEs affect deuterated systems, although to a lesser degree than hydrogenated systems. The adopted methodology is described in full detail, and simulations are, in principle, reproducible. The manuscript is well-organized and reads well, except for a few issues annotated below. Overall, this is an interesting work that may be publishable in Nature Communications, provided that the authors address the following issues:

- The title is not sufficiently descriptive; it is too broad. This work concerns NQEs on the properties of molecular liquids.
- The last paragraph on page 2 is awkward. It is nonsensical to write that density functional theory (which I would assume refers to electronic structure) does not account for quantum effects. As an electronic structure method, it is naturally quantum

mechanical (for electron) and relies on the Born-Oppenheimer approximation: Nuclei are neither classical nor quantum; they do not move. It is not strictly necessary to have a "forcefield that describes the BO surface": Ab initio PIMD has been done already in the late 1990s and early 2000s (see works by Tuckerman, Marx, Car, Parrinello, including Ref. 1), but of course, it is computationally expensive. For example, NQEs in water have been explored extensively (Markland et al. Chem. Rev. 2016).

- The forcefield (TAFFI) used to describe the interactions in the liquids should be mentioned by the end of the introduction, adding a brief explanation of the advantages of this approach over standard forcefields such as GAFF or OPLS and limitations compared to higher-level methods.

- NQEs on hydrogen-bonded systems are very sensitive to the details of the interatomic potentials and often have counterbalancing effects. In ice and water, for example, quantum delocalization effects nearly cancel out, leading to a very small variation of the melting point. I did not find any validation of the forcefield used in this manuscript against experimental measurements or higher-level quantum chemical calculations that would convince me that the subtle interplay of NQEs in H-bonded liquids is captured correctly. The statement, "These systematic increases signify weaker cohesive intermolecular interactions due to nuclear delocalization, consistent with prior studies on water and other liquid organic systems," may be expanded with quantitative comparisons.

- Page 4: "a null deuteration effect" is probably "a FULL deuteration effect".

- Page 5, line 143: the word "substances" appears to be misplaced.

- I do not fully grasp the purpose of the section "Detailed physics of linear versus branched systems". While the relations among branching, hydrogen bonding, and thermal expansion are interesting, NQEs are not discussed explicitly.

Version 1:

Reviewer comments:

Reviewer #1

(Remarks to the Author)

The authors have addressed in detail the points raised in my previous report – and, in my view, the questions raised by other reviewers as well. I appreciate the analysis included for the case of water, and their finding that the authors' ML-based prediction agrees with the weak NQE observed in water at ambient conditions. I find the new version of the manuscript suitable for publication in Nat Comm.

Reviewer #2

(Remarks to the Author)

The authors have adequately addressed all my concerns, including going beyond my expectations by performing additional simulations of alkanes and including the data in the revised manuscript. As a result, I feel that the manuscript is ready for publication.

Reviewer #3

(Remarks to the Author)

I appreciate the authors' thorough and thoughtful responses regarding:

The clarification of how quantum nuclear motion influences hydrogen bonding,

The quantification of anharmonic effects, and

The correction of the vaporization enthalpy equation.

Having also considered the comments from the other referees, I have revised my opinion: I now agree that the finding—that nuclear quantum effects have a small but noteworthy effect on physical properties of organic molecular liquids at room temperature—can be of interest (and in some cases, even surprising) to a broad audience, even if similar conclusions have previously been drawn, albeit on smaller and simpler sets of molecules (e.g., Ref. 56). I also appreciate the authors' point that some of the computational techniques employed here—though I initially regarded them as overly technical—are in fact becoming standard practice in the field.

Overall, I am happy to support the publication of this manuscript in Nature Communications.

Reviewer #4

(Remarks to the Author)

The authors have responded satisfactorily to all my previous comments and have substantially improved the readability of the manuscript. This work is of very good quality and deserves to be published in Nature Communications.

RESPONSE TO REVIEWER 1;
MANUSCRIPT ID: NCOMMS-25-00617

- **Reviewer Comments:** “Plain, quoted text”
 - **Author Reply:** *Italicized font*
 - **Changes to the manuscript:** Boxed text with revised text in red
- All page locations reference the manuscript with markups shown.

“In this work, Ugur and Webb perform path-integral computer simulations of a large number (87) of liquids/molecules at ambient conditions. The aim of this work is to identify small molecules for which the thermodynamic properties of the corresponding liquid (nuclear volume, thermal expansion coefficient, isothermal compressibility, and dielectric constant, enthalpy of vaporization) are/are not sensitive to nuclear quantum effects (as well as isotope H/D substitution effects). It is found that nuclear quantum effects (NQE) at ambient conditions (1 bar, 298 K) are relevant over a wide range of molecules, inducing up to a 5%-change in the corresponding liquid molar volume. Using machine learning, the authors show that four thermodynamic properties obtained from classical MD simulations are good indicators of those liquids for which the target thermodynamic properties are/are not affected by NQE. The question raised in this work is important and the work is timely. NQE have been studied in a few molecular liquids, particularly, water. While in this case NQE can be relevant, it is not clear the impact of NQE on other small organic molecules. The manuscript is clear and well-written. While I find that the manuscript could be published in Nat Comm, there are a few points that should be addressed. ”

Author Reply: *We are grateful reviewer’s positive assessment of the work, and they have nicely articulated the central aims and conclusions of the work. We appreciate their time in reviewing our manuscript. Below, we address their specific comments and indicate any resulting modifications.*

Date: April 10, 2025.

“1) That NQE should be relevant at low temperatures is expected. However, in this work the NQE are observed at ambient conditions (1 bar and 298 K). This may be somewhat surprising since even for water, the thermodynamic properties are barely sensitive to NQE [e.g., Eltareb et al. Phys Chem Chem Phys. 23, 6914-6928 (2021); Stolte, et al., J Phys Chem Lett. 15, 12144-12150 (2024)]. Can the authors comment on this? Have the authors studied water as well? (it is hard to tell from Fig. 1A).

Related to this question, how relevant is “a $\leq 5\%$ change in molar volume”? Can the authors provide an example/comparison showing the relevance of a 5%-change in molar volume? (eg, how would the density change of the liquids?) ”

Author Reply: *Indeed, we think that highlighting the possible relevance at ambient conditions is an important facet of our study, as well as the emphasis on organic molecules, rather than water. The reviewer makes an important observation regarding water, and the point of apparent contrast deserves some commentary.*

For water, the low apparent magnitude of nuclear quantum effects on a macroscopic property, such as the molar volume, has been partly attributed to a competition of contributions. To our understanding, the low magnitude of NQEs do not indicate that they are unimportant; they highlight that their manifestation leads to a mitigated effect on the observable. This is actually emphasized quite well in the reference by Stolte et al. This mitigation is a result of weakened hydrogen bonds due to zero-point energy and quantum tunneling competing with enhanced dipole moment due to intramolecular zero-point motion as characterized by Habershon et al. (10.1063/1.3167790). The notable impact of NQEs on the dipole moment is unique to water, as nuclear delocalization of hydrogen encompasses a significant portion of the hydrogen-bonding network unlike what is observed in our set of molecules.

To address the reviewer’s comment in the present work, we have further investigated the q-TIP4P/F water system studied by Eltareb et al. and others to ascertain how well it fits with our conclusions. Interestingly, we find that the thermal expansion coefficients (α_P) from the classical MD simulation is $2.3 \times 10^{-4} K^{-1}$, which is approximately 5.5 times smaller than the average α_P from our study. This value is reasonably consistent with the experimental value at 30 °C, $3.0 \times 10^{-4} K^{-1}$. When we utilize our newly developed simple ML model and supply the simulated features for q-TIP4P/F water, we find that we can reasonably predict the low magnitude of its NQEs (ML model estimate of 0.47% vs. simulated value of 0.25% for Δ_{v_m}). Given that our ML model also predicts a low Δ_{v_m} , we do not think our conclusions are inconsistent with prior studies on water.

We have modified the text to discuss NQEs with water and how our data-driven model captures the diminished observed effect for the molar volume.

Results, Page 4: For example, in water, anharmonicity in the O–H bond contributes to competing effects: intermolecular zero-point energy and tunneling weaken hydrogen bonding, while intramolecular zero-point motion enhances the dipole moment and strengthens interactions, resulting in relatively small net NQEs.^[44]

Results, Page 7: Although variable importance was established within the context of 92 organic molecules, it is useful to consider how these principles translate to a well-studied system, like water. Using classical simulation data from the q-TIP4P/F water model^[44] as input, the data-driven model predicts $\hat{\Delta}_{v_m}^{\text{RF}} = 0.47\%$, in good agreement with the simulation result $\Delta_{v_m} = 0.25\%$. Despite water being chemically distinct from the other molecules studied, the model captures the small impact on v_m due to NQEs—an outcome largely driven by its low α_P .

To address the reviewer’s comment on relevance of molar volume, we included a direct comparison of changes in molar volume (Δ_{v_m}) with those in density (Δ_ρ) for all systems (Suppl. Fig. 1). We now stated in the main text that $\Delta_\rho \approx -\Delta_{v_m}$. We also added a brief description on the relevance of molar volume to other material properties in page 5.

Supplementary Information, Page S2:

S1. Comparison of NQEs on Molar Volume and Density

To evaluate how nuclear quantum effects on molar volume may connect to effects on density, we compare their magnitudes across the 92 studied molecules.

Supplementary Figure 1: Comparison of magnitude of NQEs on material density and molar volume. The change in densities, Δ_ρ , is equal to $-\frac{\Delta v_m}{1-\Delta v_m}$. At low Δv_m values, Δ_ρ and $-\Delta v_m$ are equivalent, as evident in the comparison of each value across the 92 studied systems. Most significant deviation occurs for methanol with the highest magnitude of NQEs.

Results, Page 4: For reference, trends in density are opposite but similar in magnitude to molar volume, as analytically $\Delta_\rho = -\frac{\Delta_{v_m}}{1-\Delta_{v_m}} \approx -\Delta_{v_m}$ (Suppl. Fig. 1).

Results, Page 5: Given its consistent and statistically resolvable effects, in addition to v_m being a fundamental thermophysical property relevant to equations of state and other material properties, our discussion focuses on Δ_{v_m} , while analyses of other properties are included in Suppl. Fig. 3.

“2) It is not clear how/why the 87 molecules studied were chosen. Is there any chemical feature or protocol followed to select them? Even when 87 molecules is a relevant number of molecules, it is a small sample of chemical space (eg, in Fig. 2A, it seems that the region of low- Z_1 is not well-covered by the selected 87 molecules).”

Author Reply: *The 87 organic molecules studied in this work were the subset of molecules chosen in a study by Coleman et al. (10.1021/ct200731v); they were also the basis of investigation for developing the force field employed in our study (TAFFI) (10.1021/acs.jcim.1c00491). The justification is thus mostly literature continuity and the availability of experimental data, which was used for benchmarking. This set does possess many common substances of everyday relevance. We have provided additional commentary in the manuscript to explain the rationale and origin of these 87 molecules.*

Results, Page 3: To understand how NQEs manifest across chemical space, we simulate 92 organic molecules at ambient conditions. Of the 92 molecules studied, 87 were previously simulated using classical MD with various force fields,⁵¹ including TAFFI.⁵⁰ Benchmarking against experimental thermophysical properties showed that TAFFI compared favorably to other widely used empirical force fields in terms of reproducing experimental thermophysical properties across this set, which features amines, ethers, nitriles, ketones, alcohols, aldehydes, esters, amides, as well as some sulfur- and halogen-containing compounds.

With respect to the limited representation in certain areas of chemical space, we have supplemented the original 87 with an additional five molecules that are within the parameterized scope of TAFFI. These molecules were chosen to population the low Z_1 region in the UMAP space and features longer linear hydrocarbons. Linear hydrocarbons have some limited prior work in studies regarding NQEs (10.1073/pnas.1806064115), making their inclusion more relevant. We updated our analyses based on these additional data points as evident in Fig. 2A. These additions did not affect prior trends or conclusions.

Results, Page 3,4: We further augmented this set with five n-alkanes (C_9 , C_{10} , C_{11} , C_{14} , and C_{15}) to have representation of nonpolar hydrocarbons.

Figure 2A, Page 23:

“3) The use of machine learning to identify molecules prompt to NQE at ambient conditions is interesting. Unfortunately, the results are obscured by the numerical/statistical algorithms and lack of details. In page 4, the UMAP algorithm is used to classify the molecules in the Z1-Z2 plane. The algorithm should be described briefly. The collective variables Z1 and Z2 should be defined. (Fig. 2A). Similarly, the SHAP algorithm should also be described briefly in page 6.”

Author Reply: *Thanks for the recommendation. The utilization of UMAP/unsupervised learning is mostly a means of visualization across the dataset. That being the case, it is important to provide better explanation and technical detail for the benefit of readers regarding these techniques and our intent behind them. We included a brief description of the UMAP and SHAP algorithms in the main text, including a statement on the meaning of the collective variables Z_1 and Z_2 . Additional technical details of each algorithm is included in the Methods section. In addition, we have supplied notebooks on a GitHub that illustrate how to generate our analyses/figures.*

Results, Page 5,6: To visualize how NQEs vary across a broad chemical space, we apply dimensionality reduction from a high-dimensional molecular feature space to two dimensions using unsupervised machine learning. Specifically, we use the Uniform Manifold Approximation and Projection (UMAP) algorithm^{[65][66]} on a dataset that includes our 92 investigated molecules and an extended set of 2,874 molecules from the ChEMBL database.^{[67][68]} Each molecule is initially represented by a 34-dimensional atom-type feature vector generated using the Merck Molecular Force Field (MMFF94),^[69] and then UMAP constructs a neighborhood graph from the collection of such vectors, models pairwise relationships probabilistically, and optimizes a low-dimensional embedding that preserves both local and global structure. The resulting collective variables, Z_1 and Z_2 , are non-linear transformations of the high-dimensional feature values, which obfuscates facile interpretation. Nevertheless, Z_1 and Z_2 define a coordinate space that can be easily visualized and for which chemical similarity trends with distance.

Results, Page 6:

We use Shapley Additive Explanations (SHAP) analysis^[71] to attribute feature contributions to predictions made by our machine learning model. This technique assesses the impact of each feature by computing the difference in model output with and without its inclusion for all feature combinations.

“4) Can the authors use their statistical results to identify some chemistry principle (e.g., by pointing to chemical groups) that give rise to NQE? Is/isn’t there a correlation between the chemical groups and the presence of NQE in the studied molecules? The topic is briefly addressed in pages 5-6 in the context of “hydrogen bonding and branching” but it should be expanded.”

Author Reply: *Indeed, we discuss specific functional groups on population-level effects in page 6 and refer to Suppl. Fig. 5, where we explicitly examine individual functional groups. We expanded this discussion based on the reviewer’s suggestions. At present, we believe that our observations related to the properties that can predict the magnitude of NQEs further provide useful chemical insight. As our conclusions highlight the importance of features that implicitly account for intermolecular interactions (n_{H}^{cl} , α_{P}) it is a challenging task to model NQEs purely based on chemical structure. We also acknowledge that more clear attribution to functional groups and interactions is worthwhile for future study, likely requiring a substantially expanded dataset.*

Results, Page 6: In terms of readily identifiable trends, chemical structures displaying lower $\Delta_{v_{\text{m}}}$ are seemingly clustered in the Z_1 - Z_2 space in the vicinity of molecules with generally higher molecular weights, such as chloroform. In contrast, molecules with larger $\Delta_{v_{\text{m}}}$ are distributed across the manifold, indicating that diverse molecular features may enhance NQEs. This is further supported by the fact that different functional groups can be associated with population-level effects on $\Delta_{v_{\text{m}}}$ (Suppl. Fig. 5). For instance, molecules with amines tend to exhibit higher $\Delta_{v_{\text{m}}}$, and those containing heavier atoms exhibit lower $\Delta_{v_{\text{m}}}$. At the same time, there is significant variation of $\Delta_{v_{\text{m}}}$ within each population of functional groups, such that attributing NQEs purely to functional groups is difficult. This highlights the necessity to consider additional factors to ultimately understand the manifestation of NQEs.

“5) In Fig.1B and 1C, what is the meaning of the colored cloud in the panels? I understand that the dispersion of the cloud along the y-axis represents the dispersion of the NQE effects $\Delta\lambda$. However, what is the dispersion of the cloud along the x-axes? Are units needed in the x-axes? Do the x-axes represent normalized properties?”

Author Reply: *We appreciate the opportunity to clarify. We are using a data representation method often references as a “violin” plot, which is an alternative to scatter plots with error bars or box-and-whisker plots (which here we actually embed within the violins). The shape of the violins is actually an approximation to the entire distribution of data, produced through kernel density estimation. The width (spread along the x-axis at a given y value) is proportional to the frequency/density of observations for the value y. It is effectively a probability distribution but mirrored to produce the violin shape. This is particularly useful when the data may not comport to a Gaussian distribution. Thus, in this case, the shapes illustrate the distribution of $\Delta\lambda$ and $\Delta\lambda_{D\rightarrow H}$ values. The white dot represents the median, the black box represents the interquartile range, and the inner lines represent the 1.5x interquartile range. Units are not needed in the x-axes as the dispersion is normalized. We have added the detailed descriptions to the caption under Figure 1.*

RESPONSE TO REVIEWER 2;
MANUSCRIPT ID: NCOMMS-25-00617

- **Reviewer Comments:** “Plain, quoted text”
 - **Author Reply:** *Italicized font*
 - **Changes to the manuscript:** Boxed text with revised text in red
- All page locations reference the manuscript with markups shown.

“In this manuscript, Ugur and Webb using path integral molecular dynamics (PIMD) simulations to examine nuclear quantum effects (NQE) across a wide range of molecular liquids at ambient conditions, finding that NQEs are significant across the range of liquids. By computing thermodynamic quantities and performing further analysis using both classic techniques and machine learning, the authors identify physically meaningful trends that can be interpreted in terms of intuitive and chemically meaningful information. The complex data is presented in a clear and intuitive manner that helps identify trends and the major points being made. Overall, this is an interesting paper that should have broad appeal, and I recommend publication after the authors consider a few questions/comments.”

Author Reply: *We are very appreciate of the reviewer’s assessment and positive endorsement of the work. Addressing their comments have further allowed use to strengthen the manuscript, and we have responded point-by-point below.*

Date: April 10, 2025.

“1.) I think the comment “a null deuteration effect does not necessarily indicate negligible NQEs” is important. Nothing to do here, just want to point at that this is a useful statement. On a side note, the sentence needs a period at the end.”

Author Reply: *We thank the reviewer for their positive comment. We have added a period at the end of the sentence.*

“2.) On line 135, the authors mention excluding methanal, but prior to this, they do not point out that methanal is the outlier. Could you please point this out before this sentence? It is also not clear to me why having a low boiling point would cause problems. Could you please explain this point in a little more detail?”

Author Reply: *We very much appreciate the push on this comment, which motivated us to investigate the situation for methanal more closely. Initially, we were inclined to attribute its “outlier” behavior to the fact that its physical properties are different from the balance of our data. In particular, methanal exhibited a large thermal expansion coefficient, which forced the data-driven model to effectively make an “out-of-domain” prediction, prompting us to treat it as an outlier. However, with some additional insight and testing, we have created a new data-driven model that effectively captures the behavior of methanal, and even water (which was not in the scope of study). This was achieved by using a random forest model to predict a scaled quantity using n_{H} and m_{w} and then accounting for the contribution of α_{P} separately. Remarkably, this approach brings the methanal predictions virtually in line with all others. As such, we have eliminated the discussion regarding methanal being an outlier, as it is no longer required, and the results are more satisfying. Introducing the new data-driven modeling approach has required revisions in its presentation and discussion as shown below.*

Results, Page 6: To gain insight into what factors correlate with NQEs, we create a data-driven model for $\Delta_{v_{\text{m}}}$ from simple descriptors, with the rationale that an effective model would highlight the importance of those descriptors. We find that a model using just three inputs—average atomic mass (m_{w}), hydrogen density (n_{H}^{cl}), and thermal expansion coefficient ($\alpha_{\text{P}}^{\text{cl}}$)—accurately predicts $\Delta_{v_{\text{m}}}$ (Figure 2B). The coefficient of determination over all predictions is $R^2 = 0.881 \pm 0.002$. Here, predictions are made as $\hat{\Delta}_{v_{\text{m}}} = \hat{T}_{v_{\text{m}}} \alpha_{\text{P}}^{\text{cl}}$, where $\hat{T}_{v_{\text{m}}} = \text{RF}(m_{\text{w}}, n_{\text{H}}^{\text{cl}})$ is the output of a random forest (RF) regressor trained to predict a defined quantity $T_{v_{\text{m}}} \equiv (\alpha_{\text{P}}^{\text{cl}})^{-1} \Delta_{v_{\text{m}}}$, and the RF model is trained on all data except that of the molecule being predicted. We empirically find that this approach outperforms models formulated as $\hat{\Delta}_{v_{\text{m}}} = \text{RF}(m_{\text{w}}, n_{\text{H}}^{\text{cl}}, \alpha_{\text{P}}^{\text{cl}})$, likely due to the large variability in $\alpha_{\text{P}}^{\text{cl}}$ across molecules and its nature as a response function, unlike the more intrinsic descriptors m_{w} and n_{H}^{cl} . As a point of possible interest, we note that $T_{v_{\text{m}}}$ has units of temperature and, to leading order, can be interpreted as the temperature shift required for a classical system to match the molar volume of its quantum counterpart (i.e., $v_{\text{m}}^{\text{cl}}(T + T_{v_{\text{m}}}) \approx v_{\text{m}}^{\text{PI}}(T)$). This interpretation does not imply a true physical equivalence between the quantum system and a classical system at elevated temperature—a comparison that has been cautioned against as potentially misleading in prior work.^[70] Across the molecules studied, $T_{v_{\text{m}}}$ ranges from a few to several tens of degrees (Suppl. Fig. 6).

Figure 2B, 2C, Page 23:

Figure 2: Correlation of chemical features with the magnitude of NQEs. (A) A two-dimensional manifold organization of the 92 organic molecules (blue, larger markers) and 2,874 small molecules (tan, smaller markers) obtained from the ChEMBL database. The manifold is generated using the Uniform Manifold Approximation and Projection (UMAP) unsupervised learning algorithm using the 34-dimensional Merck molecular force field (69) (MMFF94) atom typing as the input feature vector. Inset lines highlight representative molecules from different regions of the manifold. (B) Comparison of predicted versus simulated effect of NQEs on molar volume (Δ_{vm}). Predicted values are obtained from our data-driven model with three chemically interpretable input features: average atomic mass (m_w), hydrogen density (n_H^{cl}), and thermal expansion coefficient (α_P^{cl}), where the latter two are determined from classical MD simulations. Error bars reflecting the standard error of the mean, determined using four independent simulations and from five different training and testing cycles for RF data, are generally smaller than the symbol size. (C) Impact of feature contributions to random forest predictions based on Shapley Additive Explanations (SHAP) analysis. The position on the x-axis indicates the impact of each feature on the model output, and the marker color indicates feature value. Feature values are transformed using a Yeo-Johnson power transformation followed by min-max scaling, such that all features are on the same scale. (78) For visual clarity of trends, the SHAP data is shown over a restricted range; Suppl. Fig. 4 depicts the full observed range of data.

“3.) Line 143 needs “to” between “due” and “the”, and a space between “substances” and n_H^{cl} .”

Author Reply: *We thank the reviewer for detecting the typographical errors. They have been rectified.*

“4.) The work seems to focus on polar liquids and/or liquids with specific interactions. Do you have any expectations for how nonpolar liquids like alkanes or fluorinated hydrocarbons would fit into the new perspective provided here? For example, would alkanes not follow the trends involving n_H ?”

Author Reply: *It is a good question. The apparent emphasis on polar liquids was not necessarily intentional. Our rationale for the 87 “chosen” molecules was mostly based on literature precedence and availability of experimental data. A group of shorter alkanes has also been examined in work by others (10.1073/pnas.1806064115), allowing us to place some additional context on our results. To address the question, we supplemented our data with five additional nonpolar alkanes with 9, 10, 11, 14, and 15 carbon atoms. We find that our model accurately predicts Δ_{vm} for these alkanes, and their placement in Figure 3 is in line with our observations framed around α_P and n_H . Our study also goes beyond focusing on specific*

interactions and probes the effect of various functional groups on population-level NQEs in page 6, as referred to in Suppl. Fig. 5. We would anticipate that fluorinated hydrocarbons would follow the same trends dictated by α_P and n_H , but it is something that would need to be confirmed in future study with adequate force fields.

Results, Page 3,4: We further augmented this set with five n-alkanes (C₉, C₁₀, C₁₁, C₁₄, and C₁₅) to have representation of nonpolar hydrocarbons.

Figure 2A, Page 23:

Figure 3, Page 24:

“5.) Line 554 has a space in the middle of the word “simulations” that should be removed.”

Author Reply: We thank the reviewer for pointing out this error.

RESPONSE TO REVIEWER 3;
MANUSCRIPT ID: NCOMMS-25-00617

- **Reviewer Comments:** “Plain, quoted text”
 - **Author Reply:** *Italicized font*
 - **Changes to the manuscript:** Boxed text with revised text in red
- All page locations reference the manuscript with markups shown.

“I enjoyed reading this paper, which explores local and global molecular indicators of quantum nuclear effects (NQEs) in molecular liquids. The analysis is sophisticated, employing state-of-the-art machine learning approaches. Their method successfully captures intuitive indicators of NQEs, such as those associated with the presence of light atoms. However, beyond these expected trends, I did not find the results entirely convincing.”

Author Reply: *We are happy to hear the reviewer enjoyed the paper. Although retrospectively we also find many of the trends to be intuitive, others may be less so, such as consideration of thermal expansion coefficient as an indicator of NQEs, which we have not seen explicated elsewhere. In addition, we would suggest that the observation of substantive NQEs at ambient conditions for numerous organic liquids is something that some may find surprising, as evidenced by other referee comments, for example. We will address the latter comment about whether the results are convincing momentarily as we discuss additional data.*

Date: April 18, 2025.

“- Some observed trends may even contradict previous studies. For example, Ref. [1] demonstrates that stronger hydrogen bonds enhance NQEs in the Ubbelohde effect, whereas this paper suggests that weaker interactions lead to stronger hydrogen bonding. ”

Author Reply: *We do not fully agree with this characterization of our results (which we accept as a possible limitation of the writing) or that our results are in any apparent contradiction with the literature regarding the importance of hydrogen bonding. For example, in Figure 3, we do observe an overall positive trend with hydrogen density, which would have a connection with hydrogen bonding. Moreover, we also highlight that hydrogen bonding enhances NQEs with specific comparisons of molecules lacking hydrogen-bonding groups (1-bromobutane, 1-chlorobutane, butane-1-thiol) with butan-1-ol in Figure 4A, and discuss the “positive impact of hydrogen bonding on n_H and Δ_{v_m} ” on page 7.*

What we do reveal (by comparing molecules with one, two, and three hydroxyl groups) is that that presence of multiple hydrogen-bonding group does not necessarily enhance NQEs because it tends to decrease α_P . This also highlights an important consideration raised by our work, which is that fluid stability, which we associate with the thermal expansion coefficient, is a conflating factor when making comparison between molecules. When you compare molecules with similar α_P , we again observe the general enhancement of NQEs with increased hydrogen density. So, our suggestion is that our analysis expands the scope of considerations that may be relevant for manifestation of NQEs. We have sought to clarify some discussion in the manuscript that may be interpreted as discounting the importance of hydrogen bonding.

Results, Page 7, 8: However, n_H and Δ_{v_m} do not scale monotonically with additional hydrogen-bonding groups. This is revealed by comparison of butan-1-ol (iv), butane-1,4-diol (v), pentane-1,5-diol (vi), and propane-1,2,3-triol (vii). Among these alcohols, increasing the number of hydrogen-bonding groups minimally affects hydrogen density but increases the stability of the fluid (decreases α_P).

“- Could this discrepancy arise because the force field used lacks anharmonicity?

- NQEs are known to be significantly modulated by anharmonic effects, which the authors do not account for [1,2,3]. At a minimum, this limitation should be explicitly discussed.”

[1] <https://www.pnas.org/doi/10.1073/pnas.1016653108>

[2] <https://journals.aps.org/prl/abstract/10.1103/PhysRevLett.117.115702>

[3] <https://pubs.acs.org/doi/10.1021/acs.jctc.9b00596>

Author Reply: *As per the earlier discussion, we do not necessarily believe there is a discrepancy, per se. Nevertheless, it is true that the force field, in its stretching and bending intramolecular terms, does not directly account for anharmonicity in the parameterization. To ascertain whether this has significant impact on our results, we reevaluated the magnitude of NQEs in a subset of 29 molecules with hydrogen bonding groups, performing simulations with stretching terms fit to a function that can capture anharmonicity in the oscillation. In particular, we performed DFT scans of hydroxyl, amine, and thiol bond stretches at the ω B97X-D3/def2-TZVP level of theory. We then described the resulting energy functions with the anharmonic Morse potential as done in the q-TIP4P/F water model (10.1063/1.3167790).*

Overall, we find that the anharmonicity of these modes has a relatively small impact on the magnitude of NQEs. For most molecules, including anharmonicity slightly suppresses observed NQEs, which is somewhat expected, but not by enough to change the major trends presented in the original manuscript.

We speculate that the notable effect of anharmonicity on water is a direct result of the significant changes in the dipole moment of the molecule. In contrast, hydrogen bonding groups constitute a much smaller part of our molecules, and their anharmonicity does not result in significant changes in the dipole moments. We have added our explicit discussion of anharmonicity in the main text and included additional simulation results in Suppl. Fig. 2. We incorporated the references provided by the reviewer in the main text. We also added a statement in page 5 indicating that anharmonicity may play an important role and merits extra care for other systems in future studies.

Results, Page 4, 5: For the original TAFFI framework, all bonds are fit to purely harmonic functions, but anharmonicity in the potential energy surface—especially in bond-stretching modes—has been identified as a key factor influencing the magnitude of NQEs.⁴⁴ ⁶² ⁶⁴ For example, in water, anharmonicity in the O–H bond contributes to competing effects: intermolecular zero-point energy and tunneling weaken hydrogen bonding, while intramolecular zero-point motion enhances the dipole moment and strengthens interactions, resulting in relatively small net NQEs.⁴⁴

To evaluate then whether the harmonic description in TAFFI may be responsible for the large NQEs relative to water, we re-parameterized bonds involving hydroxyls, amines, and thiols (29 molecules) with an anharmonic Morse potential⁴⁴ based on expanded mode scans at ω B97X-D3/def2-TZVP level of theory. We then evaluated Δ_{v_m} with this anharmonic bond description and compared it the harmonic results (Suppl. Fig. 2). Overall, we find that anharmonicity in bond-stretching slightly suppresses NQEs, though the overall magnitudes remain comparable to the harmonic description. We speculate that this mitigating effect is weaker in the organic molecules studied than in water, likely due to smaller dipole moment changes and correspondingly less enhancement of intermolecular interactions following introduction of anharmonicity. Intuitively, as most of the molecules in this study are larger than water, they are expected to exhibit smaller overall changes in dipole moment due to anharmonicity, as local bond distortions contribute less to the total molecular polarization. Although anharmonicity appears to have little impact for the molecules and conditions studied, its role in other phases, molecules, or force fields warrants careful consideration.

Supplementary Information, Page S3:

S2. Effect of Anharmonicity on Δ_{v_m}

As anharmonicity may play an important role in the magnitude of NQEs, we investigate systems with hydrogen bonding groups where their hydroxyl, amine, and thiol bond stretches are described by the anharmonic Morse function. The simulated magnitudes of NQEs are compared with the results from the original TAFFI force field.

Supplementary Figure 2: Effect of bond stretch anharmonicity on NQEs. Energy functions from DFT scans of hydroxyl, amine, and thiol bond stretches at the ω B97X-D3/def2-TZVP level of theory were fitted to the anharmonic Morse potential for 29 molecules with hydrogen bonding groups. The calculated magnitude of NQEs on the molar volume are displayed against the results from the original TAFFI force field with harmonic description of the bond stretch.

Results, Page 5: Although anharmonicity appears to have little impact for the molecules and conditions studied, its role in other phases, molecules, or force fields warrants careful consideration.

“- Additionally, the paper is highly computational in focus, which may limit its appeal to experimentalists. I have two technical comments that the authors should consider, especially if submitting to a computationally oriented journal:”

Author Reply: *We understand the comment. We contend that considering this suite of molecules and their behavior at ambient conditions also significantly expands the conventional scope of study for NQEs. This, along with the explicit consideration of the equilibrium isotope effect relevant to many experimental studies, should render the paper more broadly interesting than just to simulators. Furthermore, we have effectively distilled our predictive models to just a handful of quite interpretable quantities, which should enhance the appeal. There is a fair criticism to be made about the utilization of data-driven analysis and how that may be received or interpreted from more of a layman perspective. While we generally expect that the trend in science to be overall growing literacy in application of such machine learning methods, we have made efforts to better explain these methods. That being said, we have added brief descriptions of the methods used for analysis to increase the accessibility of our manuscript to a broader audience.*

Results, Page 5,6:

To visualize how NQEs vary across a broad chemical space, we apply dimensionality reduction from a high-dimensional molecular feature space to two dimensions using unsupervised machine learning. Specifically, we use the Uniform Manifold Approximation and Projection (UMAP) algorithm^[65, 66] on a dataset that includes our 92 investigated molecules and an extended set of 2,874 molecules from the ChEMBL database.^[67, 68] Each molecule is initially represented by a 34-dimensional atom-type feature vector generated using the Merck Molecular Force Field (MMFF94),^[69] and then UMAP constructs a neighborhood graph from the collection of such vectors, models pairwise relationships probabilistically, and optimizes a low-dimensional embedding that preserves both local and global structure. The resulting collective variables, Z_1 and Z_2 , are non-linear transformations of the high-dimensional feature values, which obfuscates facile interpretation. Nevertheless, Z_1 and Z_2 define a coordinate space that can be easily visualized and for which chemical similarity trends with distance.

Results, Page 6:

We use Shapley Additive Explanations (SHAP) analysis^[71] to attribute feature contributions to predictions made by our machine learning model. This technique assesses the impact of each feature by computing the difference in model output with and without its inclusion for all feature combinations.

“- The expression for the enthalpy of vaporization is incorrect, as it assumes that the kinetic energy cancels between the liquid and gas phases.”

Author Reply: *We thank the reviewer for noting this. The equation was described imprecisely. In the original manuscript, we presumed cancellation of the classical kinetic energy parts, but the quantum kinetic energy part was included – but the equation did not reflect this (lumping it into the potential energy term). For clarity, we have reevaluated the enthalpy of vaporization to simply consider all components to use all components to the internal energy. Equation 7 and its description has been corrected based on these modifications.*

Author Reply:

Methods, Equation 7, Page 19:

Molar heats of vaporization, Δh_{vap} , were calculated using

$$\Delta h_{\text{vap}} = \frac{1}{P} \sum_{i=1}^P E_i^{(g)} + k_{\text{B}}T - \frac{1}{P} \sum_{i=1}^P E_i^{(l)} \quad (7)$$

where $E^{(\nu)}$ is the total internal energy per mole (including both potential and kinetic energy) in phase ν , either liquid or gas. This expression assumes the ideal gas approximation and that the vapor-phase molar volume is much greater than that of the liquid. Often, the kinetic energy contributions in the gas and liquid phases are expected to cancel, allowing Δh_{vap} to be approximated from potential energies alone. 51 However, in PIMD simulations, quantum kinetic energy—arising from harmonic coupling of systems in the extended phase space—can differ between phases. Therefore, we retain all energy contributions in the calculation for simplicity.

RESPONSE TO REVIEWER 4;
MANUSCRIPT ID: NCOMMS-25-00617

- **Reviewer Comments:** “Plain, quoted text”
 - **Author Reply:** *Italicized font*
 - **Changes to the manuscript:** Boxed text with revised text in red
- All page locations reference the manuscript with markups shown.

“The manuscript by Ugur and Webb reports on a high-throughput study of nuclear quantum effects in 87 organic molecular liquids at ambient conditions using path integral molecular dynamics simulations. An exhaustive analysis of NQEs on several properties of the liquid (density, compressibility, thermal expansion, etc.) is carried out using state-of-the-art supervised and unsupervised statistical learning approaches, which reveal general trends. In systems with high hydrogen density and lower stability, NQEs have a higher impact on their properties. These observations provide useful indications to assess whether treating NQEs by PIMD is necessary to study certain systems. Additionally, the simulations indicate that NQEs affect deuterated systems, although to a lesser degree than hydrogenated systems. The adopted methodology is described in full detail, and simulations are, in principle, reproducible. The manuscript is well-organized and reads well, except for a few issues annotated below. Overall, this is an interesting work that may be publishable in Nature Communications, provided that the authors address the following issues:”

Author Reply: *We thank the reviewer for their comments and insights. Below, we address their specific comments and indicate any resulting modifications.*

“- The title is not sufficiently descriptive; it is too broad. This work concerns NQEs on the properties of molecular liquids.”

Author Reply: *Fair point, we agree. We have changed the title to “Nuclear Quantum Effects in Molecular Liquids Across Chemical Space” to be more specific.*

“- The last paragraph on page 2 is awkward. It is nonsensical to write that density functional theory (which I would assume refers to electronic structure) does not account for quantum effects. As an electronic structure method, it is naturally quantum mechanical (for electron) and relies on the Born-Oppenheimer approximation: Nuclei are neither classical nor quantum; they do not move. It is not strictly necessary to have a “forcefield that describes the BO surface”: Ab initio PIMD has been done already in the late 1990s and early 2000s (see works by Tuckerman, Marx, Car, Parrinello, including Ref. 1), but of course, it is computationally expensive. For example, NQEs in water have been explored extensively (Markland et al. Chem. Rev. 2016).”

Author Reply: *We first clarify: our statement is that density functional theory alone does not account for nuclear quantum effects for the very reason highlighted by the reviewer: it is an electronic structure method. “Solving” this problem in the field of fixed nuclei thus provides the potential energy surface that is most suitable for PIMD, as emphasized by Markland et al. (10.1038/s41570-017-0109). While it may not be strictly necessary, the use of the BO PES is implied by the formalism (starting with a statement of the partition function being written as a trace of the operator $e^{-\beta\hat{H}}$). Our main point is that empirically calibrated force fields intrinsically account for NQEs (albeit in uncontrolled ways), which may result in “double counting” of NQEs, as also discussed by Habershon et al. (10.1063/1.3167790) and Kühne et al. (10.1080/00268976.2014.981231). Performing ab initio PIMD with a high quality theory would be the preferred approach for all systems. However, at the present time, as the reviewer points out, this is not currently feasible. To handle the scope of molecules featured in our study, our approach is to use an analytical potential energy surface but fitted only to quantum chemical calculations. We have modified this paragraph to include more details and ensure clarity, as highlighted below.*

Introduction, Page 2,3: Path-integral molecular dynamics (PIMD) enables explicit treatment of NQEs and calculation of isotope effects. This is in contrast to conventional classical MD, which treats nuclei as point particles evolving on a potential energy surface. Meanwhile, electronic structure methods, like density functional theory, provide a quantum mechanical treatment of electronic degrees of freedom but generally rely on the Born-Oppenheimer approximation, under which the nuclei are held fixed or propagated classically. PIMD incorporates NQEs via a mathematical isomorphism, mapping each quantum nucleus to a classical ring polymer, thereby capturing quantum delocalization and thermal fluctuations by sampling in an extended phase space [43]. For water, PIMD has demonstrated how intramolecular zero-point motion and intermolecular tunneling affect properties like translational diffusion and orientational relaxation rates. [44] [45] PIMD has also revealed the significance of NQEs in DNA base pairs, acetylene:ammonia co-crystals, and electrolyte transport in confined aqueous systems. [46] [48] Nevertheless, in addition to the computational overhead of the extended phase space, PIMD can be practically limited by the availability of force fields that accurately represent the Born-Oppenheimer potential energy surface. Most conventional classical MD simulations typically use efficient force fields based on analytical equations. However, these force fields have parameters often calibrated to experimental data, implicitly including NQEs in uncontrolled ways, such that their use in PIMD risks “double counting” NQEs. [44] [49] As a result, broad application of PIMD remains limited, and the role of NQEs across many systems and conditions is still largely unexplored.

“- The forcefield (TAFFI) used to describe the interactions in the liquids should be mentioned by the end of the introduction, adding a brief explanation of the advantages of this approach over standard forcefields such as GAFF or OPLS and limitations compared to higher-level methods.”

Author Reply: *Thanks for the recommendation. This also likely clarifies our position raised by the prior comment. We now explicitly mention TAFFI and its relevance to the study. We also explicitly compare two empirically calibrated force fields, OPLS-AA and OpenFF, with TAFFI in the context of NQEs for a subset of 11 molecules displayed in Suppl. Section 13, Suppl. Fig. 13 and 14. We find that while relative trends amongst molecules are roughly preserved, both empirical force fields tend to overestimate the magnitude of NQEs compared to TAFFI. Relative to GAFF and OPLS, TAFFI has actually been explicitly compared across a variety of properties in its original paper; all force fields are roughly comparable over this set of 87 organic molecules across the range of thermophysical properties studied. This is to suggest that TAFFI is not notably worse by comparison to widely used force fields for describing organic molecules. However, very importantly, TAFFI is parameterized entirely from quantum chemical calculations at the ω B97X-D3/def2-TZVP level of theory. As alluded to in our prior response, the fact that the non-bonded parameters are not adjusted by empirical calibration is important, in our view, for suitability in assessing NQEs by the path-integral formulation.*

Introduction, Page 3: Nevertheless, in addition to the computational overhead of the extended phase space, PIMD can be practically limited by the availability of force fields that accurately represent the Born-Oppenheimer potential energy surface. Most conventional classical MD simulations typically use efficient force fields based on analytical equations. However, these force fields have parameters often calibrated to experimental data, implicitly including NQEs in uncontrolled ways, such that their use in PIMD risks “double counting” NQEs.44 49 As a result, broad application of PIMD remains limited, and the role of NQEs across many systems and conditions is still largely unexplored.

Introduction, Page 3:

This investigation spans a diverse set of chemistries—including amines, ethers, ketones, alcohols, alkanes, and more—and is enabled by the Topology Automated Force-Field Interactions (TAFFI) framework,50 an efficient analytical force field parameterized solely from quantum chemical calculations at the ω B97X-D3/def2-TZVP level. Because TAFFI does not utilize experimental calibration, it is well-suited for conducting PIMD across the numerous systems studied.

Methods, Page 20:

As previously discussed, TAFFI is used because it relies purely on quantum chemical calculations for its parameterization and computational efficiency; however, we acknowledge that the reported magnitude of NQEs could differ based on the underlying force field.

TAFFI does, like many other classical force fields, does approximate the potential energy surface using prescribed analytical equations with limitations (strictly pairwise non-bonded interactions, harmonic potential, etc.). Ideally, we would have force fields for all molecules that are parameterized similarly to the MB-pol water model from Paesani and coworkers (accounting for many-body interactions and using a high-quality of theory).

“- NQEs on hydrogen-bonded systems are very sensitive to the details of the interatomic potentials and often have counterbalancing effects. In ice and water, for example, quantum delocalization effects nearly cancel out, leading to a very small variation of the melting point. I did not find any validation of the forcefield used in this manuscript against experimental measurements or higher-level quantum chemical calculations that would convince me that the subtle interplay of NQEs in H-bonded liquids is captured correctly. The statement, “These systematic increases signify weaker cohesive intermolecular interactions due to nuclear delocalization, consistent with prior studies on water and other liquid organic systems,” may be expanded with quantitative comparisons.”

Author Reply: *Indeed, the reviewer is correct in that effects may be PES-dependent. We have attempted to address our utilization of TAFFI in several ways.*

First, we show that the relative trends among our investigated molecules are mostly preserved via discussion and data related to Suppl. Section 13, “Impact of Force Field on Δ_{v_m} ”. This data was originally included in the original manuscript.

Second, we have directly investigated implications of anharmonicity by reparameterizing hydroxyl, amine, and thiol bond stretches of 29 molecules with at least one hydrogen-bonding group using an expanded set of DFT calculations (ω B97X-D3/def2-TZVP), fitting to an anharmonic Morse potential. This choice of function was motivated by the success of its use in the q-TIP4P/F water model by Habershon et al. (10.1063/1.3167790) that allows capturing the subtle competing effects. For these molecules, we find that anharmonicity tends to somewhat suppress NQEs, but not to an extent that affects the key observations of the original manuscript. These calculations and accompanying discussion are now also a portion of the revised manuscript (shown below).

Results, Page 4, 5: For the original TAFFI framework, all bonds are fit to purely harmonic functions, but anharmonicity in the potential energy surface—especially in bond-stretching modes—has been identified as a key factor influencing the magnitude of NQEs. [44] [62-64] For example, in water, anharmonicity in the O–H bond contributes to competing effects: intermolecular zero-point energy and tunneling weaken hydrogen bonding, while intramolecular zero-point motion enhances the dipole moment and strengthens interactions, resulting in relatively small net NQEs. [44]

To evaluate then whether the harmonic description in TAFFI may be responsible for the large NQEs relative to water, we re-parameterized bonds involving hydroxyls, amines, and thiols (29 molecules) with an anharmonic Morse potential [44] based on expanded mode scans at ω B97X-D3/def2-TZVP level of theory. We then evaluated Δ_{v_m} with this anharmonic bond description and compared it the harmonic results (Suppl. Fig. 2). Overall, we find that anharmonicity in bond-stretching slightly suppresses NQEs, though the overall magnitudes remain comparable to the harmonic description. We speculate that this mitigating effect is weaker in the organic molecules studied than in water, likely due to smaller dipole moment changes and correspondingly less enhancement of intermolecular interactions following introduction of anharmonicity. Intuitively, as most of the molecules in this study are larger than water, they are expected to exhibit smaller overall changes in dipole moment due to anharmonicity, as local bond distortions contribute less to the total molecular polarization. Although anharmonicity appears to have little impact for the molecules and conditions studied, its role in other phases, molecules, or force fields warrants careful consideration.

Supplementary Information, Page S3:

S2. Effect of Anharmonicity on Δ_{v_m}

As anharmonicity may play an important role in the magnitude of NQEs, we investigate systems with hydrogen bonding groups where their hydroxyl, amine, and thiol bond stretches are described by the anharmonic Morse function. The simulated magnitudes of NQEs are compared with the results from the original TAFFI force field.

Supplementary Figure 2: Effect of bond stretch anharmonicity on NQEs. Energy functions from DFT scans of hydroxyl, amine, and thiol bond stretches at the ω B97X-D3/def2-TZVP level of theory were fitted to the anharmonic Morse potential for 29 molecules with hydrogen bonding groups. The calculated magnitude of NQEs on the molar volume are displayed against the results from the original TAFFI force field with harmonic description of the bond stretch.

Third, while it was not prominently featured in our manuscript, it is worth mentioning that TAFFI models were robustly and favorably compared to experimental data for numerous thermophysical properties in its original paper. In this manuscript, we explicitly evaluate this in Suppl. Fig. 10 and allude to it on page 17 of the main text.

Methods, Page 17: The force field in tandem with aforementioned settings enables accurate reproduction of liquid-phase experimental densities⁵¹ for 87 systems with available empirical data (Suppl. Fig. 10). Benchmarking of classical and PIMD simulation results also shows that path-integral treatment of most systems brings predicted molar volumes closer to experiment. To gain quantitative insight, we apply a one-tailed paired t-test on the differences with experiment and report $p = 4.8 \times 10^{-17} < 0.05$, indicating that PIMD results are systematically closer to experiment. These results, along with good alignment between force-field intramolecular normal-mode frequencies and DFT results, suggest that TAFFI can effectively represent and discriminate the chemically diverse set of systems in this study.

Fourth, we better describe the use of TAFFI to provide insight into its expected quality. Parameters for TAFFI targeted reproduction of energies obtained using ω B97X-D3/def2-TZVP and showed good correspondence with intramolecular mode frequencies as an aggregate test of quality—these are results of its original paper.

Introduction, Page 3: This investigation spans a diverse set of chemistries—including amines, ethers, ketones, alcohols, alkanes, and more—and is enabled by the Topology Automated Force-Field Interactions (TAFFI) framework,⁵⁰ an efficient analytical force field parameterized solely from quantum chemical calculations at the ω B97X-D3/def2-TZVP level. Because TAFFI does not utilize experimental calibration, it is well-suited for conducting PIMD across the numerous systems studied.

Fifth, we also explored the use of a general purpose machine learning potential, MACE-OFF23, which is trained to reproduce the energies and forces computed at the ω B97M-D3(BJ)/def2-TZVPPD level of quantum mechanics. Despite the considerable computational expense (PIMD for a 1500-atom system for 0.4 ns took over 12 days), errors in MACE-OFF23 relative to experimental properties were rather large, and so we were not inclined to trust these calculations more than those produced by TAFFI. At this stage, these results are merely for the interest of the reviewer. Likely, the MLPs would need some additional fine-tuning to be more reliable. We are hoping to explore this area further in the future.

“- Page 4: “a null deuteration effect” is probably “a FULL deuteration effect”.”

Author Reply: *Thanks for the comment – the sentence was unclear. We are remarking that a small isotope effect does not necessarily imply that NQEs are insignificant, because the impact on the deuterated system versus the normal systems can partially cancel. We revised the text accordingly to avoid any ambiguity.*

Results, Page 5: This highlights that deuteration does not fully replicate a purely classical treatment. A system with a negligible isotope effect may still display significant NQEs, which partially cancel between the isotopically normal and deuterated systems.

“- Page 5, line 143: the word “substances” appears to be misplaced.”

Author Reply: *Thanks – it has been corrected.*

“- I do not fully grasp the purpose of the section “Detailed physics of linear versus branched systems”. While the relations among branching, hydrogen bonding, and thermal expansion are interesting, NQEs are not discussed explicitly.”

Author Reply: *We are happy to clarify and modify the manuscript accordingly. In this section, we aim to understand NQEs from the standpoint of chemical constitution (or more at a molecular-level). Because we have already established how NQEs are impacted by α_P and n_H , our analysis effectively examines how molecular factors (such as branching) impact the aforementioned properties (that correlate with NQEs). The text has been modified to more explicitly call attention to the trends with NQEs.*

Results, Page 8: The conventional rationale for the trend involving boiling temperatures is that branched molecules have reduced surface area and less efficient packing, which weakens intermolecular forces and impacts the magnitude of NQEs. Intriguingly, the linear and branched molecules studied here exhibit comparable interaction patterns with respect to the number of hydrogen bonds, their strength, and average nearest-neighbor distances (Suppl. Fig. 9). This renders the importance of hydrogen-bonding groups on trends in α_P , and thus impacts on Δ_{v_m} , unclear.

Results, Page 8:

These nuanced changes in intermolecular interactions due to branching result in a pronounced increase in α_P , which tends to enhance Δ_{v_m} . Notably, the minimum interaction energy for butan-1-ol occurs at a distance where few or no hydrogen bonds form. Additionally, butan-1-ol lacks a strong preference for relative molecular orientation, unlike 2-methylpropan-2-ol (Figure 5C). These findings suggest that the cohesive energy of 2-methylpropan-2-ol relies more heavily on hydrogen bonding whereas butan-1-ol exhibits overall stronger dispersion forces. This comparison also serves to demonstrate the sensitivity of NQEs to subtle changes in nanoscale interactions.